# Soft corrugated channel with synergistic exclusive discrimination gating for $CO_2$ recognition in gas mixture

Yifan Gu [1,2], Jia-Jia Zheng[3], Ken-ichi Otake [2] ✉, Shigeyoshi Sakaki[2], Hirotaka Ashitani[4], Yoshiki Kubota [4,5], Shogo Kawaguchi [6], Ming-Shui Yao[2], Ping Wang[2], Ying Wang [1], Fengting Li[1] ✉ & Susumu Kitagawa [2] ✉

Developing artificial porous systems with high molecular recognition performance is critical but very challenging to achieve selective uptake of a particular component from a mixture of many similar species, regardless of the size and affinity of these competing species. A porous platform that integrates multiple recognition mechanisms working cooperatively for highly efficient guest identification is desired. Here, we designed a flexible porous coordination polymer (PCP) and realised a corrugated channel system that cooperatively responds to only target gas molecules by taking advantage of its stereochemical shape, location of binding sites, and structural softness. The binding sites and structural deformation act synergistically, exhibiting exclusive discrimination gating (EDG) effect for selective gate-opening adsorption of $CO_2$ over nine similar gas molecules, including $N_2$, $CH_4$, CO, $O_2$, $H_2$, Ar, $C_2H_6$, and even higher-affinity gases such as $C_2H_2$ and $C_2H_4$. Combining in-situ crystallographic experiments with theoretical studies, it is clear that this unparalleled ability to decipher the $CO_2$ molecule is achieved through the coordination of framework dynamics, guest diffusion, and interaction energetics. Furthermore, the gas co-adsorption and breakthrough separation performance render the obtained PCP an efficient adsorbent for $CO_2$ capture from various gas mixtures.

Molecular recognition, one of the essential processes in chemical and biological systems, is the specific interaction between multicomponent molecular mixtures through non-covalent bonds for guest identification and selective binding[1,2]. Natural biological hosts (e.g., proteins and enzymes) typically perform the most efficient molecular recognition, which relies on their conformational dynamical complimentary dimensions with abundant binding interactions[3]. Integrating molecular recognition regimes into porous materials can help create materials with exceptional separation, selective transportation, catalysis, and sensing properties[4–6]. Although significant progress has been made, achieving the selective binding with only a target guest from multicomponent mixtures in artificial porous materials,

[1]College of Environmental Science and Engineering, State Key Laboratory of Pollution Control and Resource Reuse, Tongji University, Siping Road 1239, 200092 Shanghai, China. [2]Institute for Integrated Cell-Material Sciences (WPI-iCeMS), Kyoto University Institute for Advanced Study, Kyoto University, Yoshida Ushinomiya-cho, Sakyo-ku, Kyoto 606-8501, Japan. [3]Laboratory of Theoretical and Computational Nanoscience, National Center for Nanoscience and Technology, Chinese Academy of Sciences, 100190 Beijing, China. [4]Department of Physical Science, Graduate School of Science, Osaka Prefecture University, Sakai, Osaka 599-8531, Japan. [5]Department of Physics, Graduate School of Science, Osaka Metropolitan University, Sakai, Osaka 599-8531, Japan. [6]Japan Synchrotron Radiation Research Insitute (JASRI), SPring-8, 1-1-1 Kouto, Sayo-cho, Sayo-gun, Hyogo 679-5198, Japan. ✉e-mail: ootake.kenichi.8a@kyoto-u.ac.jp; fengting@tongji.edu.cn; kitagawa@icems.kyoto-u.ac.jp

regardless of the sizes and affinities of other competing species, is still particularly challenging. To develop such an intelligent molecular recognition mechanism, porous structures utilising sophisticated supramolecular interactions should be actively explored[7–9].

Porous coordination polymers (PCPs) or metal-organic frameworks (MOFs) are highly designable porous materials for constructing efficient recognisers[10–17]. Similar to zeolite and mesoporous silica, rigid PCPs can exhibit size-exclusion or diffusion-limited effects by tuning pore size and shape to sort different-sized molecules (Fig. 1a)[18–22]. The spatial organisation of interacting sites within PCPs contributes to their recognition properties by boosting target guests' binding affinities (Fig. 1b)[6,23–28]. However, these PCPs show unsatisfactory selectivity for multicomponent mixtures of guests, which are smaller than the pore apertures or those with similar affinities. Recently, flexible PCPs exhibiting guest-triggered reversible structural transformations have attracted significant attention[29–33]. This structural transformation, called 'gate-opening behaviour,' endows soft PCPs with specific molecular discrimination capability[34–37]. The subtle energy balance

among the thermodynamic interactions involving guest molecules and the framework deformation energy can explain this selective gate-opening mechanism (Fig. 1c)[38–41]. However, it is also challenging to selectively uptake guests with a weaker affinity to the soft PCP over ones with a higher affinity. Furthermore, the co-adsorption of undesired guests in flexible PCPs after gate opening is also problematic[42].

We propose that synergistically utilising all these available recognition mechanisms through manipulating the energetic and stereochemical features of PCPs may bring unprecedented permeable recognition performance, for example, the specific selective adsorption of gases with moderate affinity from a mixture of components with similar structure and properties. In reality, such recognition property is highly desired in separation systems for mixtures, specifically for capturing only the target impurity molecules from a multi-component mixture through a single energy-efficient adsorption step. Herein, we demonstrate that a soft PCP featuring an adaptable, narrow-corrugated channel is a versatile platform (Fig. 1d) for intelligent molecular recognition through the exclusive discrimination gating

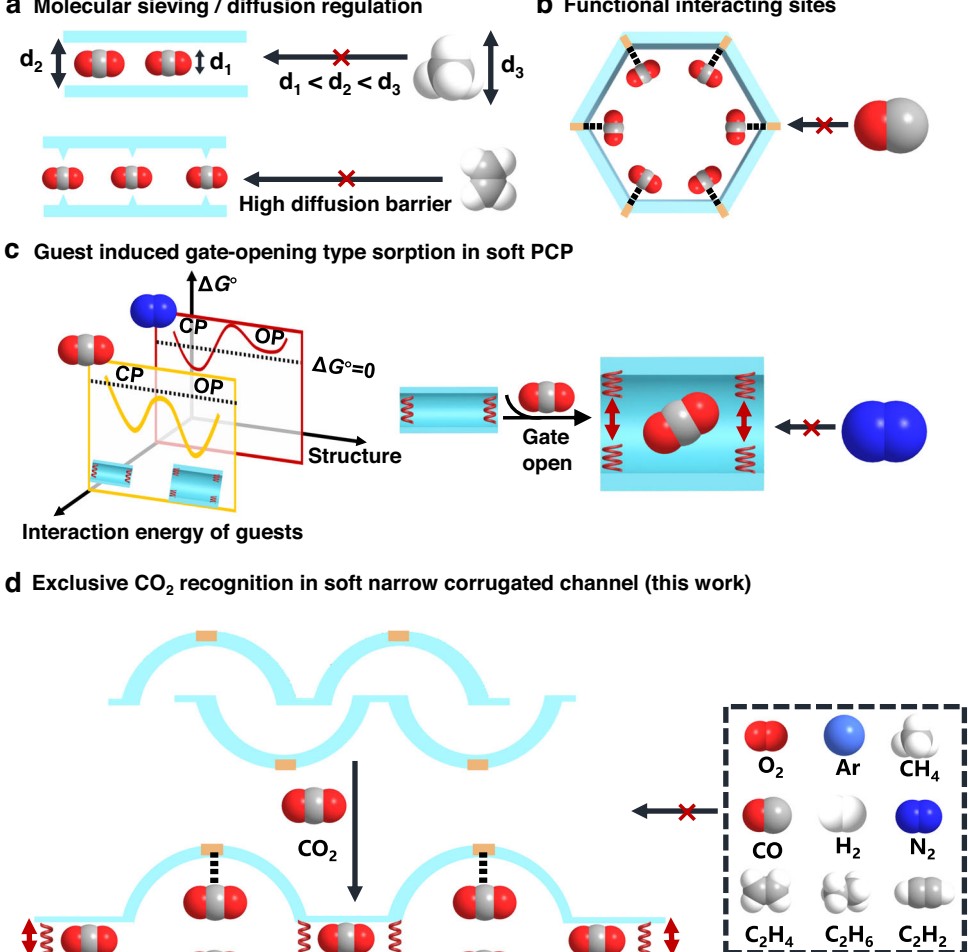

**Fig. 1 | Exclusive molecular recognition mechanisms in PCPs. a** Rigid PCPs can exhibit size-exclusion or diffusion-limited effect for guest recognition. $d_2$ is the size of the channel, which is between the size of the target gases ($d_1$) and the size of the rest ($d_3$). **b** Functional interacting sites within PCPs for guest recognition. The incorporated binding sites show a more robust interaction with the target gases than the rest. **c** In a soft PCP system, guest adsorption accompanied by structural

transformations from closed phase (CP) to open phase (OP) occurs only when the interaction energy surpasses the structural deformation energy between the PCP and gas molecules. **d** PCPs with exclusive discrimination gating (EDG), which can efficiently utilise all the above mechanisms synergistically, exhibit precise molecular recognition performance (flexible framework with restricted narrow-corrugated channel structure is a suitable candidate).

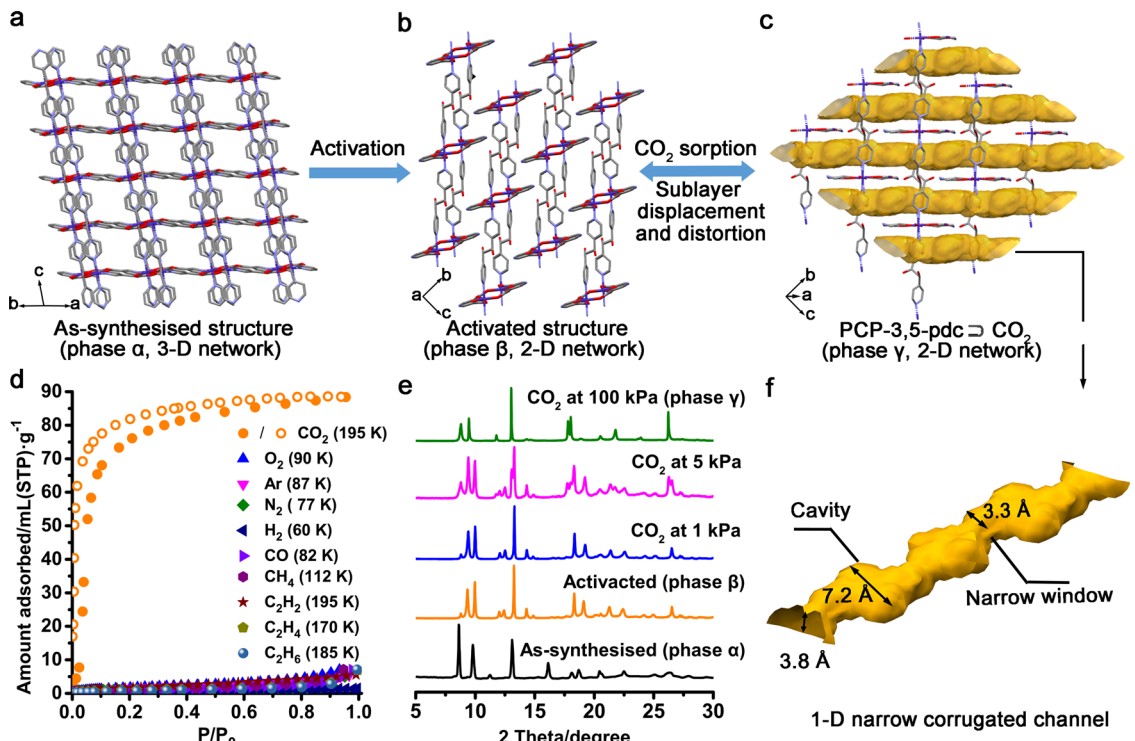

**Fig. 2 | Framework flexibility and sorption profiles of PCP-3,5-pdc. a** Overall framework of the as-synthesised 3-D network structure of PCP-3,5-pdc (phase α). **b** Overall framework of the activated 2-D layer structure of PCP-3,5-pdc (phase β). **c** $CO_2$ adsorption induced structural transformation of PCP-3,5-pdc. The accessible 1-D channels in PCP-3,5-pdc (phase γ) are highlighted in yellow (Probe radius: 1.2 Å). Purple, red, blue, and grey in the PCP frameworks represent Co, O, N and C, respectively. The hydrogen atoms and guest molecules are omitted for clarity. **d** Adsorption isotherms of $N_2$, $CO_2$, $CH_4$, CO, $O_2$, $H_2$, Ar, $C_2H_2$, $C_2H_4$, and $C_2H_6$ at low temperatures. The solid and open circles in red represent the adsorption and desorption of $CO_2$, respectively. **e** In-situ synchrotron PXRD patterns of PCP-3,5-pdc accompanying $CO_2$ adsorption at 195 K. **f** A view of the 1-D narrow-corrugated channel in phase γ structure of PCP-3,5-pdc.

(EDG) effect. As a 'smart gate,' the dynamic bottleneck aperture in the corrugated channel can adjust its size and shape to regulate guest diffusion. Simultaneously, it can also thermodynamically distinguish the target gas from other competing gases with fewer host-guest interactions. Additionally, the interacting sites on the cavity surface in a corrugated channel can provide further specific interactions with a target molecule, synergistically improving recognition capability. In this research, a flexible 2-D interdigitated framework with relatively strong metal-ligand coordination bonds was designed to provide suitable pore geometry and structural softness for selective $CO_2$ adsorption. $CO_2$ is an essential greenhouse gas and a significant impurity in a mixture of various industrial gases[43,44]. Benefiting from the collaboration of the binding sites and soft corrugated-channel structures, this PCP exhibits a unique exclusive discrimination gating (EDG) effect for $CO_2$ over the competing nine similar gaseous molecules, including $N_2$, $CH_4$, CO, $O_2$, $H_2$, Ar, $C_2H_6$, and even gases with similar molecular size but higher affinity, such as $C_2H_4$ and $C_2H_2$ (Supplementary Table 1). Such novel molecular recognition capability from the proposed structural design strategy yields porous materials with high performance in challenging recognition and separation systems.

## Results and discussion
### Crystal structures and gases sorption selectivity
Pink-coloured single crystals of $[Co(3,5-pdc)dpg]_n$ (PCP-3,5-pdc; 3,5-pdc= 3,5-pyridinedicarboxylic acid; dpg=meso-α,β-di(4-pyridyl) glycol) were synthesised via the solvothermal reaction of $Co(NO_3)_2 \cdot 6H_2O$, dpg, and 3,5-pdc in mixed DMF/MeOH solutions (Supplementary Figs. 1 and 2). Single-crystal X-ray diffraction (SXRD) analysis revealed that as-synthesised PCP-3,5-pdc crystallised in the monoclinic I2/a space group (Supplementary Table 2). In the as-synthesised structure

phase α, the Co centre was a complex with six coordinates exhibiting an octahedral geometry. Each Co(II) ion coordinates with three dpg ligands and three 3,5-pdc ligands to form a 3-D network (Fig. 2a and Supplementary Fig. 3). Activation of the as-synthesised samples at 120 °C under vacuum led to a change in the PXRD pattern, indicating the formation of guest-free PCP-3,5-pdc (phase β, Fig. 2b). Thermo-gravimetric analysis (TGA) demonstrated that the guest molecules were removed entirely and that the coordination framework was thermally stable up to ca. 270 °C (Supplementary Fig. 4). SXRD analysis of activated PCP-3,5-pdc revealed that the coordination environment of the Co(II) centre and global framework underwent a drastic recasting and distortion process (Supplementary Table 3). During the activation process, the original coordination bonds between the Co centre and N of the 3,5-pdc ligand and the O atom of the hydroxyl group of the dpg ligand were broken. Meanwhile, coordination bonds between the Co centre and the original uncoordinated O atom from the carboxylic group of the 3,5-pdc ligand were formed (Supplementary Fig. 5). Such rearrangement of the coordination bonding geometry enables the structural transformation of the framework from a 3-D network to a 2-D system (Fig. 2b). In phase β, each Co(II) ion is in a distorted octahedral geometry coordinated by two dpg ligands at the axial positions, two oxygen atoms from the chelating carboxylate end of the 3,5-pdc ligand, and two oxygen atoms from 3,5-pdc in the equatorial plane (Supplementary Fig. 3). Co(II) and 3,5-pdc give rise to 1-D double-chain structures along the axis, and further linkages of these chains via dpg ligands in the axial positions generate a dense interdigitated 2-D layer structure (Supplementary Fig. 6). The purity of phases α and β were confirmed by comparing the simulated and experimental synchrotron powder X-ray diffraction (PXRD) patterns (Supplementary Fig. 7). Coordination polymers with interdigitated (CID) 2-D sheet systems have long been studied as good motifs for

dynamic structures[31,45]. The constituent sublayer networks may dislocate their mutual positions upon specific gas adsorption, thereby increasing the adequate pore size. Therefore, the structural nature of activated PCP-3,5-pdc may result in exciting gas adsorption and recognition behaviour.

To investigate the adsorption properties of the presented PCP, molecular probes of 10 small molecular gases, including $N_2$, $CH_4$, $CO$, $CO_2$, $O_2$, $H_2$, $Ar$, $C_2H_2$, $C_2H_4$, and $C_2H_6$ were utilised. Unprecedently, the prepared PCP exclusively exhibited $CO_2$ recognition. As shown in Fig. 2d, e, the uptake below the gate-opening pressure is attributed to the inclusion of $CO_2$ in the intrinsic microporous cavities of phase β (Supplementary Table 4 and Supplementary Fig. 8). Then, the gate-opening behaviour was observed at the pressure of $P/P_0 = 0.05$ at 195 K, leading to final maximum adsorption of 90 mL·g$^{-1}$ at 1 bar (corresponding to 1.75 molecules per $Co^{2+}$). The Hill coefficient ($n$), which is a measure of the degree of adsorption cooperativity for the gate-opening step in the $CO_2$ adsorption isotherm at 195 K, was determined to be 4.1 (>1) (Supplementary Fig. 9), confirming the positive, cooperative adsorption phenomenon. Conversely, negligible adsorption of $N_2$, $CH_4$, $CO$, $O_2$, $H_2$, $Ar$, $C_2H_2$, $C_2H_4$, and $C_2H_6$ was observed at low temperatures, indicating that these nine gases could not unlock the open-phase structure of PCP-3,5-pdc. Notably, distinguishing between $C_2H_2$ and $CO_2$ is one of the most challenging tasks due to the similarities in their boiling points (194.7 K for $CO_2$ and 189.3 K for $C_2H_2$) and kinetic diameters (3.3 Å for both molecules)[46]. Additionally, the higher quadrupole moment of $C_2H_2$ ($7.2 \times 10^{-26}$ esu cm$^2$) compared to that of $CO_2$ ($0.65 \times 10^{-26}$ esu cm$^2$) often results in more vital electrostatic interaction of $C_2H_2$ with the adsorbent, causing preferential adsorption of $C_2H_2$ over $CO_2$[47]. This inverse selectivity of $CO_2$ over $C_2H_2$ harvested in this PCP is relatively rare. Isobar adsorption measurements further confirmed that the other nine gases, except $CO_2$, did not induce the gate-opening behaviour of PCP-3,5-pdc (Supplementary Figs. 10 and 11).

Even under high-pressure conditions (Supplementary Fig. 12), PCP-3,5-pdc adsorbed $CO_2$ selectively. As illustrated in Supplementary Fig. 13, the compound offered a two-step adsorption isotherm for $CO_2$ at 25 bar and 298 K. The uptake in the first step (20 mL·g$^{-1}$) is attributed to the inclusion of $CO_2$ into the microporous cavities of PCP-3,5-pdc (phase β). The second step of $CO_2$ uptake exhibits a characteristic sigmoidal hysteretic adsorption, which suggests gate opening in PCP-3,5-pdc, leading to a final maximum uptake of ca. 55 mL·g$^{-1}$. The positive $CO_2$ adsorption cooperativity in the structural transformation step was even stronger (Hill coefficient $n = 6.6$, Supplementary Fig. 14). The difference in cooperativity at different temperatures may be due to the varying diffusion and stabilising abilities of the $CO_2$ molecules. At the same time, no apparent gate-opening type adsorption was observed for $N_2$, $CH_4$, $CO$, $O_2$, $H_2$, $Ar$, $C_2H_4$ and $C_2H_6$ at 298 K. Notably, the slopes of the $C_2H_4$ and $CO_2$ adsorption isotherms before the gate-opening pressure (Supplementary Fig. 15) indicated that PCP-3,5-pdc (phase β) exhibits stronger $C_2H_4$-framework interactions than $CO_2$ before the transformation of the framework structure. Even so, the lack of further gate-open adsorption indicates that $C_2H_4$ cannot provide the open structure of PCP-3,5-pdc. These gas adsorption results demonstrate the unprecedented $CO_2$ molecular recognition capability of PCP-3,5-pdc.

High-pressure mixture gas co-adsorption (Supplementary Fig. 16) and breakthrough separation (Supplementary Fig. 17) tests were conducted at room temperature to evaluate further the $CO_2$ recognition performance of PCP-3,5-pdc in mixed-gas systems. As shown in Fig. 3a–c, the mixed-gas adsorption of $CO_2/N_2$, $CO_2/CH_4$, and $CO_2/C_2H_4$ (50.0/50.0 v/v) at 298 K was measured up to a total pressure of 20 bar for PCP-3,5-pdc. The volume of $N_2$, $CH_4$, and $C_2H_4$ adsorbed in the binary adsorption mixtures was 0.5, 2.8 and 3.3 mL·g$^{-1}$, respectively. Still, the amount of $CO_2$ adsorbed was similar to that observed in the single-component experiments,

indicating that PCP-3,5-pdc maintained high selectivity for $CO_2$ in gas mixtures. Notably, unlike $N_2$ and $CH_4$, $C_2H_4$ adsorption shows a higher uptake than $CO_2$ in the low-pressure region (single gas pressure < 1 bar), indicating that PCP-3,5-pdc shows stronger interactions with $C_2H_4$ than $CO_2$ in the initial state (Supplementary Fig. 18). Even so, only $CO_2$ adsorption can further induce the gate opening of PCP-3,5-pdc and can be selectively adsorbed into the open framework at high pressure. These results suggest that the adsorption of $CO_2$ in the open framework of PCP-3,5-pdc is strong enough to block the pore entry, preventing effective guest exchange in the pore network and resulting in high adsorption selectivity. The separation factors ($S$) calculated from the co-adsorption isotherms were up to 116.6, 16.0, and 10.9 $CO_2/N_2$, $CO_2/CH_4$, and $CO_2/C_2H_4$, respectively (Fig. 3d), suggesting promising potential of PCP-3,5-pdc for trapping $CO_2$ under dynamic conditions. To verify this hypothesis, a mixture of $N_2/CO_2$, $CH_4/CO_2$ or $C_2H_4/CO_2$ (50:50, v/v) at a flow rate of 6 mL/min was passed through a fixed-bed column filled with activated PCP-3,5-pdc at room temperature. Remarkable $CO_2$ capture performance was achieved for the $N_2/CO_2$ and $CH_4/CO_2$ mixtures (Supplementary Figs. 19 and 20). Pure $N_2$ or $CH_4$ gas was first eluted from the separation bed, whereas $CO_2$ was captured as an impurity in a packed column until its saturated uptake. The separation performance of a $CO_2/C_2H_4$ binary mixture in PCP-35-pdc was relatively weaker (Supplementary Fig. 21). Unlike $N_2$ and $CH_4$, $C_2H_4$ adsorption shows a higher uptake than $CO_2$ before pore opening at room temperature, indicating that PCP-3,5-pdc shows stronger interactions with $C_2H_4$ than $CO_2$ in the initial state (Fig. 3c). This observation is consistent with the calculated binding energies of these gas molecules at different pore-opening states for this PCP. The preferential adsorption of $C_2H_4$ in the closed phase could potentially affect the separation performance of the $CO_2/C_2H_4$ mixture due to the insufficient adsorption equilibrium under dynamic conditions, in contrast to co-adsorption experiments.

An ideal adsorbent for practical applications should allow for recyclability and energy-efficient regeneration. Therefore, we performed cycling breakthrough experiments for $N_2/CO_2$ separation under the same conditions. Between each cycle, the PCPs were regenerated under an in-situ vacuum without heating. The results showed that both PCPs maintained the same retention time (Supplementary Fig. 22). Moreover, after the high-pressure adsorption tests, PCP-3,5-pdc retained its original crystal structure (Supplementary Fig. 23). PCP-3,5-pdc exhibits excellent stability making it conducive for practical applications, especially under high-pressure separation conditions, such as precombustion $CO_2$ capture (20–30 bar) and natural gas processing (50 bar)[27,43]. To test the humidity stability, we exposed PCP-3,5-pdc to 75% relative humidity at room temperature for seven days. This PCP retained its $CO_2$ adsorption capacity (Supplementary Fig. 24). In addition, the impact of humidity on the separation performance was investigated by comparing the $CO_2/N_2$ breakthrough curves of the sample exposed to humid air for more than one week, both before and after thermal activation (Supplementary Fig. 25). The results indicate that while the $CO_2/N_2$ separation ability was sustained, its performance degraded due to humidity. However, since the structure remained stable even under humid conditions, the separation performance was not affected after activation. The water moisture resistance performance of PCP may originate from the lack of strong hydrophilic sites, as suggested by the water vapour adsorption isotherm (Supplementary Fig. 26).

## Gas-loaded crystal structure

To monitor the structural transformation of PCP-3,5-pdc, we conducted in-situ synchrotron PXRD measurements during $CO_2$ adsorption at 195 K. As shown in Fig. 2e, guest-free PCP-3,5-pdc starts to transform into a new phase structure (phase γ) at a $CO_2$ pressure of $P/P_0 = 0.05$. This structural change corresponds to the sudden opening

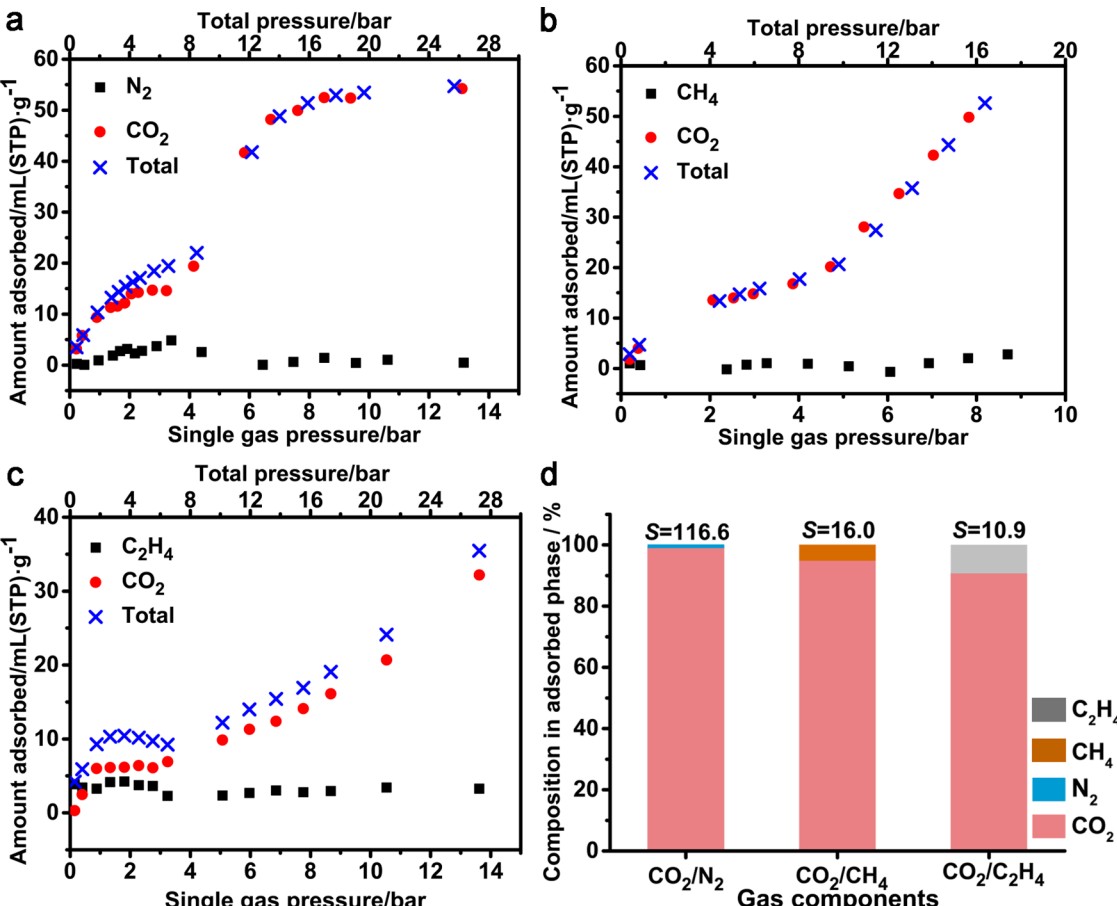

**Fig. 3 | Mixture gas co-sorption and breakthrough separation profiles.**
**a** Volumetric high-pressure co-adsorption equilibria of a binary mixture $N_2/CO_2$
(50.0/50.0 v/v) at 298 K of PCP-3,5-pdc. **b** Volumetric high-pressure co-adsorption
equilibria of a binary mixture $CH_4/CO_2$ (50.0/50.0 v/v) at 298 K of PCP-3,5-pdc.
**c** Volumetric high-pressure co-adsorption equilibria of a binary mixture $C_2H_4/CO_2$
(50.0/50.0 v/v) at 298 K of PCP-3,5-pdc. **d** Separation factors ($S$) calculated from the
co-adsorption isotherms for $CO_2/N_2$, $CO_2/CH_4$, and $CO_2/C_2H_4$ (50.0/50.0 v/v) gas
mixtures.

of the pores of the framework from a dense structure (phase β) to a
wide-pore structure (phase γ) during $CO_2$ uptake (Fig. 2c). For com-
parison, in-situ synchrotron PXRD, accompanied by $C_2H_2$ sorption at
195 K, was also recorded (Supplementary Fig. 27). As expected, PCP-
3,5-pdc retains its dense phase structure, suggesting no structural
response to $C_2H_2$ adsorption. $C_2H_2$ has a similar kinetic diameter to
that of $CO_2$. Still, the adsorption behaviour of $C_2H_2$ differs from that of
$CO_2$, suggesting that some other factors lead to the difference between
$CO_2$ and $C_2H_2$, as discussed below. The crystal structure of phase γ was
successfully determined by an ab initio charge-flipping method, and
subsequent structural refinement was performed using Rietveld ana-
lyses of in-situ synchrotron PXRD data (Supplementary Fig. 28 and
Supplementary Table 5). As shown in Fig. 2c, the open-phase structure
is formed by the distortion and displacement of the original densely
packed 2-D interdigitated sublayers in response to $CO_2$ uptake, pro-
viding accessible 1-D narrow-corrugated channels (Fig. 2c, f). These
infinite undulating channels in phase γ structure feature relatively
large cavities (7.2 × 3.8 Å) connected by ultra-small windows
(3.8 × 3.3 Å). The narrow-sized window perfectly matches the dimen-
sions of $CO_2$ for diffusion in the channel, potentially the key mechan-
ism for $CO_2$ adsorption selectivity. The in-situ synchrotron PXRD data
during $CO_2$ desorption at 195 K further indicates that the dense-to-
open phase transformation is fully reversible (Supplementary Fig. 29).
Additional in-situ PXRD measurements during $CO_2$ adsorption at 298 K
up to 10 bar further verified the corresponding elastic structure of
PCP-3,5-pdc upon $CO_2$ uptake at high pressure (Supplemen-
tary Fig. 30).

Visualisation of the binding positions of $CO_2$ was also achieved
through the Rietveld refinement of $CO_2$-loaded crystals. In the refined
crystal structure, the adsorbed $CO_2$ molecules assemble into 1-D chains
along the channel direction (Fig. 4a). These chains were stabilised by
gas-gas interactions between $CO_2$ molecules from two types of binding
sites in a T-shaped geometry [$C^{(δ+)}···O^{(δ-)}$ = 4.366 Å]. At site I, $CO_2$
molecules diffused through the narrow windows. They were adsorbed
in relatively large cavities, sandwiched by two pyrazine units of the 3,5-
pdc ligand (Fig. 4b). At site II, $CO_2$ molecules were adsorbed parallel to
the narrow windows, sandwiched by the two pyrazine units of the dpg
ligand. Such narrow window sizes can provide an unavoidable steric
hindrance for the diffusion of other gases with larger sizes, leading to a
high selectivity for $CO_2$ adsorption. The presence of the negatively
charged N atom from the 3,5-pdc ligand allows additional interactions
with $CO_2$ [$CO_2$-I, $C^{(δ+)}···N^{(δ-)}$ = 3.441 Å; $CO_2$-II, $C^{(δ+)}···N^{(δ-)}$ = 3.584 Å,
Fig. 4b], providing an electrostatic $CO_2$-framework interaction for
recognition.

**Flexible framework with a narrow-corrugated channel**
To further verify the design strategy for a 'flexible framework with a
narrow-corrugated channel' to enhance guest selectivity, we con-
ducted theoretical calculations of $CO_2$, $N_2$, $CH_4$, $C_2H_2$, and $C_2H_4$
adsorptions to PCP-3,5-pdc. The adsorption structures were obtained
by classical Monte Carlo simulation[48] followed by the geometry opti-
misation using the periodic density functional theory (DFT) calculation
with the PBE-D3 functional[49]. The calculation finds the $CO_2$ molecules
at site I in the broader cavity and site II in the narrow window, similar to

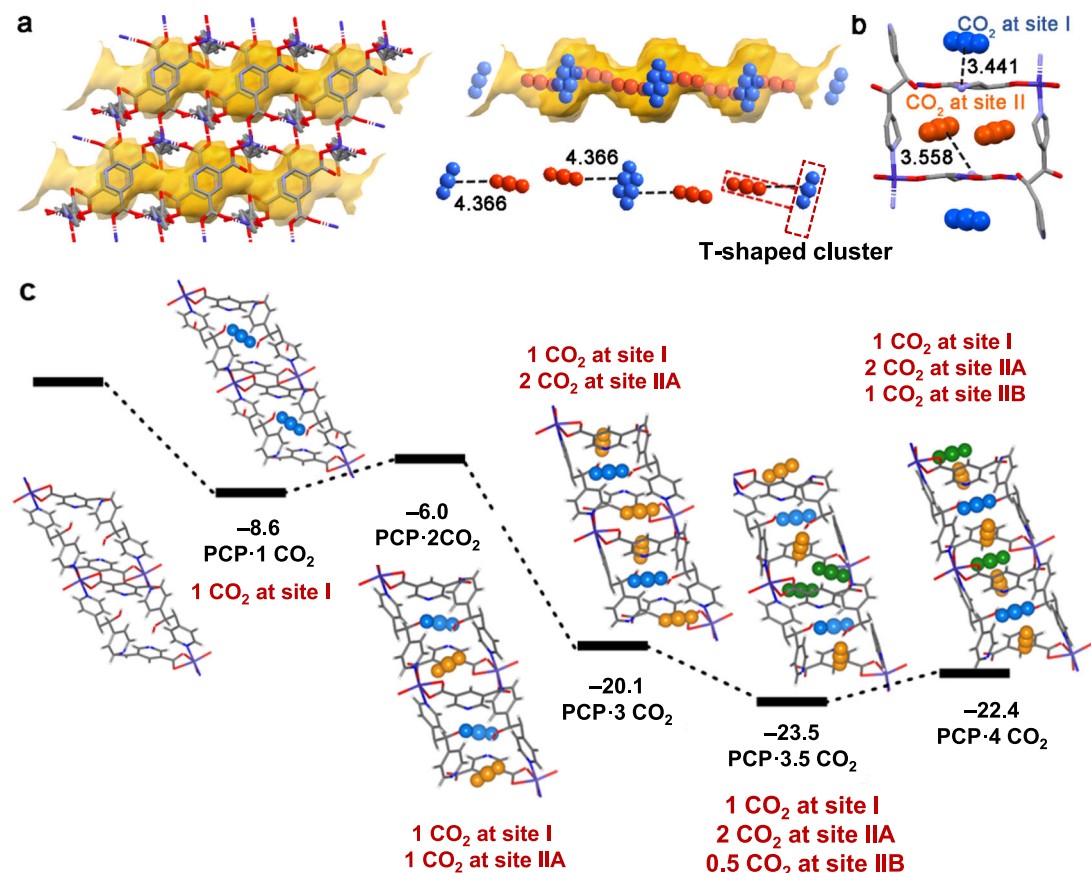

**Fig. 4 | CO₂ binding sites in PCP-3,5-pdc. a** The $CO_2$ adsorption structure in PCP-3,5-pdc, showing two types of adsorption sites ($CO_2$-I is highlighted in blue and $CO_2$-II in orange). The $CO_2$-I is shown as disordered, with occupancies of 50%. **b** $CO_2$-framework interactions. Purple, red, blue, and grey in the PCP framework represent Co, O, N and C, respectively. The hydrogen atoms of molecules are omitted for clarity. The black dashed line represents interactions, and the value along the line represents the distance (in Å). **c** DFT-calculated energy diagram for $CO_2$ adsorption into PCP-3,5-pdc, where $CO_2$-I is highlighted in blue, $CO_2$-IIA in orange and $CO_2$-IIB in green. Energies are given in kcal mol$^{-1}$.

those determined by the experiment (Supplementary Fig. 31). In site II, there exist two possible adsorption positions, named IIA and IIB. The binding energy (BE) for $CO_2$ adsorption at these sites decreases in the order of site I (−8.6 kcal mol$^{-1}$) > site IIA (−2.4 kcal mol$^{-1}$) > site IIB (−1.0 kcal mol$^{-1}$), as shown in Supplementary Table 6, where the negative value of BE means adsorption is exothermic. This result suggests that $CO_2$ molecules occupy site I first. Because the experimentally observed adsorption amount of $CO_2$ was approximately 3.5 molecules per unit cell of PCP-3,5-pdc, which corresponds to 1.75 molecules per $Co^{2+}$, we investigated the binding energies for the subsequent $CO_2$ adsorption at the sites IIA and IIB (Supplementary Table 7). As shown in Fig. 4c, the BE decreases (less negative) when the second $CO_2$ molecule is adsorbed at the site IIA (PCP·2$CO_2$) in the presence of one molecule at site I, but increases (more negative) when one more $CO_2$ molecule is simultaneously adsorbed at the site IIA (PCP·2$CO_2$) even in the presence of one molecule at the site I. The increase in BE arises from the formation of T-shaped molecular clusters of $CO_2$ between the adsorbed $CO_2$ molecules stabilising gas-gas interactions, which stabilises gas-gas interactions, and the decrease in average deformation energy of PCP induced by $CO_2$ adsorptions (Fig. 4a and Supplementary Table 7). However, the BE value decreases (less negative) when four $CO_2$ molecules are adsorbed at all the adsorption sites II to afford PCP·4$CO_2$ (Fig. 4c) because of the congestion due to the 'narrow-corrugated channel', as suggested by the decreased interaction energy ($E_{int, H-G}$) between $CO_2$ molecules and PCP framework (Supplementary Table 7). Interestingly, the total BE value is the largest (the most negative) when one $CO_2$ molecule is adsorbed at site I, two $CO_2$ molecules are adsorbed at site IIA, and one

$CO_2$ molecule is adsorbed at one site IIB of two unit cells, suggesting that simultaneous adsorption of more than one $CO_2$ molecules can happen in PCP-3,5-pdc, which is a typical feature of gate-opening adsorption[39].

The adsorption sites of other gaseous molecules are similar to those of $CO_2$ (Supplementary Fig. 31). Among these positions, the adsorption at site I is the most stable; the BE value is the most negative (Supplementary Table 6). At the adsorption site I, $N_2$ and $CH_4$ exhibit less negative BE values (−6.1 and −7.6 kcal mol$^{-1}$) than that of $CO_2$ (−8.6 kcal mol$^{-1}$), suggesting the weaker affinity of PCP-3,5-pdc to $N_2$ and $CH_4$ than to $CO_2$. This is reasonable because the interaction of $N_2$ and $CH_4$ molecules with the PCP framework is usually weaker than that of $CO_2$, probably since $N_2$ and $CH_4$ are much less polarised. This common feature is useful for selectively adsorbing $CO_2$ over $N_2$ and $CH_4$. Therefore, the exclusion of $N_2$ and $CH_4$ from the gas mixture occurs through the thermodynamic mechanism. However, we should note that the BE values for $C_2H_2$ and $C_2H_4$ adsorptions at site I (−9.9 and −11.9 kcal mol$^{-1}$, respectively) are more negative than for $CO_2$ adsorption (Supplementary Table 6). The stronger binding affinity of PCP-3,5-pdc to $C_2H_4$ was also verified by high-pressure single-gas adsorption (Supplementary Fig. 15) and mixture gas co-adsorption measurements (Supplementary Fig. 18). The results are consistent with the observation that $C_2H_4$ loading is more significant than that of $CO_2$ at high temperature before the gate-opening pressure for $CO_2$ adsorption but seemingly against our experimental results indicating that PCP-3,5-pdc can selectively adsorb $CO_2$ over $C_2H_2$ and $C_2H_4$ at low temperature. We also investigated the energy diagrams for the subsequent adsorption of $C_2H_2$ and $C_2H_4$ at site II (Supplementary Fig. 32) and found that the

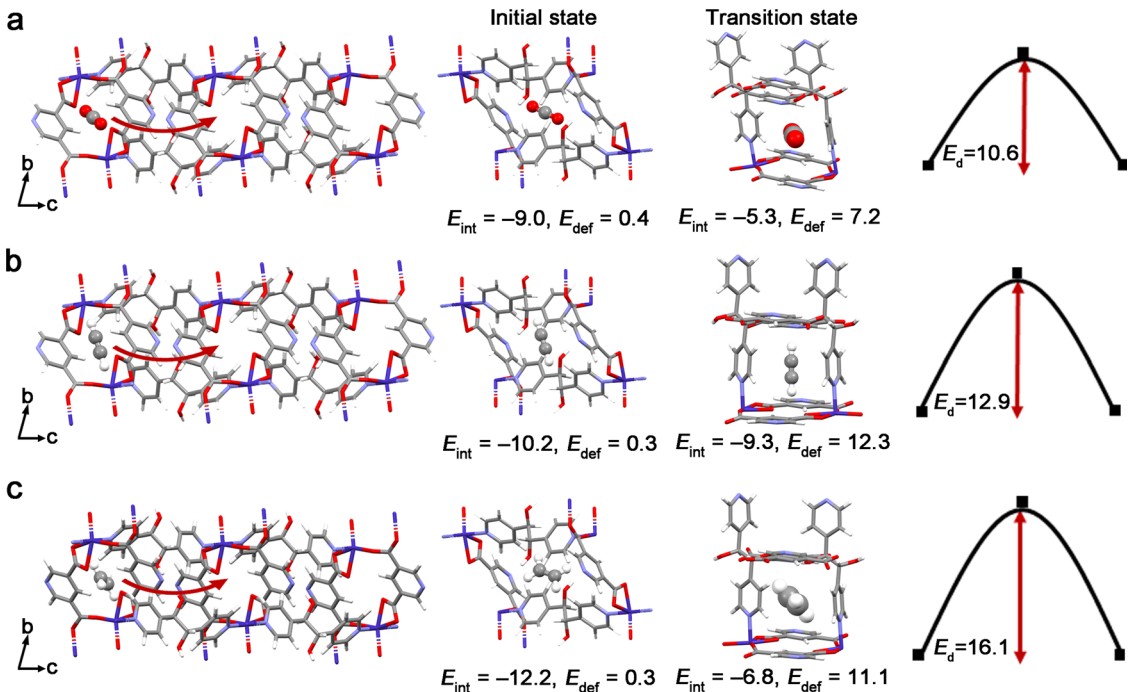

**Fig. 5 | A schematic diagram of the diffusion of $CO_2$, $C_2H_2$ and $C_2H_4$ in the narrow-corrugated channel of PCP-3,5-pdc. a** $CO_2$ molecule at the initial state (IS), transition state (TS) for the diffusion of $CO_2$ through the narrow window, and the diffusion pathway of $CO_2$ in the 1-D corrugated channel. **b** $C_2H_2$ molecule at IS, TS for the diffusion of $C_2H_2$ through the narrow window, and the diffusion pathway of $C_2H_2$ in the 1-D corrugated channel. **c** $C_2H_4$ molecule at IS, TS for the diffusion of $C_2H_4$ through the narrow window, and the diffusion pathway of $C_2H_4$ in the 1-D corrugated channel. Purple, red, blue, grey and pink in the PCP frameworks and gas molecules represent Co, O, N, C and H, respectively. The diffusion barrier was calculated using the computational model with one adsorbed gas molecule. We considered a diffusion process in which gas molecules move from site I to its neighbouring site II. Interaction energies ($E_{int}$), deformation energies ($E_{def}$), and diffusion barriers ($E_b$) are given in kcal mol$^{-1}$.

adsorption occurs similarly to that of $CO_2$, as follows: The subsequent adsorption of one more $C_2H_2$ or $C_2H_4$ at the site II leads to a decrease (less negative) in BE, but the simultaneous adsorption of more than one $C_2H_2$ or $C_2H_4$ molecules at site II results in more negative BE values. Such changes in BE suggest that gate-opening adsorption of $C_2H_2$ or $C_2H_4$ is possible. These results suggest that the selective adsorption of $CO_2$ over $C_2H_2$ and $C_2H_4$ at low temperatures was not thermodynamically but could be kinetically controlled, as discussed below.

To verify whether the kinetic factor contributes to the selective recognition of $CO_2$ over $C_2H_2$ and $C_2H_4$ or not, we calculated diffusion barriers for these gas molecules moving through the narrow-corrugated channel (Fig. 5 and Supplementary Fig. 33). The DFT-calculated activation barrier ($E_a$) for diffusion decreases in the order of $C_2H_4$ (16.1 kcal mol$^{-1}$) > $C_2H_2$ (12.9 kcal mol$^{-1}$) > $CO_2$ (10.6 kcal mol$^{-1}$), suggesting that $C_2H_4$ and $C_2H_2$ are more challenging to enter into the pore of PCP-3,5-pdc than does $CO_2$. To find the reason why the diffusion of $C_2H_2$ and $C_2H_4$ needs a larger barrier than $CO_2$, we analysed the interaction energies ($E_{int}$) and deformation energies ($E_{def}$) at the adsorption structures (i.e., the initial state of diffusion and transition state of diffusion, named IS and TS, respectively). The $E_{int}$ at the IS decreases in the order of $C_2H_4$ > $C_2H_2$ > $CO_2$, which is a standard feature for the interaction energies of these gas molecules with PCP frameworks. In addition, the $E_{def}$ is negligibly small for all the gas molecules because adsorption at site I induces minor structural deformation of the framework. As going from IS to TS, the structural expansion of the framework occurs, leading to an increase in $E_{def}$. In addition, the $E_{int}$ decreases (less negative) because the gas molecule leaves its best position for interacting with the framework. Because $C_2H_4$ has a larger size than $CO_2$, the structural expansion of the framework induced by $C_2H_4$ diffusion is larger, leading to a larger $E_{def}$ at TS for $C_2H_4$ than for $CO_2$. Thus, the diffusion of $C_2H_4$ needs a much larger barrier than that of $CO_2$. On the other hand, this feature is not

observed for comparison between $CO_2$ and $C_2H_2$ because their sizes are similar. However, these two gas molecules have different electronic structures[50] and thus interact with the framework differently. As shown in Supplementary Fig. 34, the $C_2H_2$ molecule interacts with the framework via the C-H···O interaction, where the positively charged H atom of $C_2H_2$ approaches the negatively charged O atoms of the 3,5-pdc ligand, whereas $CO_2$ does not form such an interaction. Such an interaction yields a more negative $E_{int}$ for $C_2H_2$ at both IS and TS than $CO_2$. However, to create the C-H···O interaction, the $C_2H_2$ molecule must keep its orientation to be perpendicular to the channel wall, which gives rise to larger structural deformation because that $C_2H_2$ must exist at a narrow channel at TS. As a result, the $E_{def}$ for $C_2H_2$ at TS is much larger than that for $CO_2$. Thus, despite the $E_{int}$ less decreases the $E_{def}$ increases much more at TS than at IS in the $C_2H_2$ case as going from IS to TS, leading to a larger $E_a$ in the $C_2H_2$ diffusion. These results indicate that the ideal gate size, adsorption structure (or intermolecular interaction manner), and limited framework flexibility in PCP-3,5-pdc is vital to the limited diffusion of $C_2H_2$ and $C_2H_4$. Therefore, because of the restricted flexible framework with a narrow-corrugated channel structure, PCP-3,5-pdc can achieve unusual selective adsorption of $CO_2$ through structural response to gas diffusion despite weaker binding affinity of $CO_2$ than those of $C_2H_2$ and $C_2H_4$. In other words, the selective adsorption of $CO_2$ over $C_2H_2$ and $C_2H_4$ occurs through kinetics. The measured adsorption isobars illustrate that $C_2H_2$ and $C_2H_4$ molecules can slightly enter the pores. In contrast, diffusion is facilitated at elevated temperatures (Supplementary Fig. 35). Additional analyses of $C_2H_2$ and $C_2H_4$ adsorption at different exposure times and temperatures further confirmed that the observed recognition of $CO_2$ over $C_2H_2$ and $C_2H_4$ is the result of kinetic factors (Supplementary Figs. 36–38).

We successfully demonstrated precise $CO_2$ recognition over nine similar, small gaseous molecules using a designed flexible PCP with

exclusive discrimination gating behaviour. Compared with known molecule recognition strategies (Supplementary Table 8), including molecular sieving and functional interacting sites, the optimal cooperation of stereochemical shape, location of binding sites, and structural softness through designing narrow-corrugated channel structures in soft porous materials bring unprecedented recognition efficiency, especially for obtaining the desired selectivity in multi-component mixtures containing molecules of various sizes and affinities. This strategy offers a promising blueprint for designing high-performance porous materials with high performance in challenging recognition and separation systems.

# Methods

## Materials

Cobalt nitrate hexahydrate ($Co(NO_3)_2 \cdot 6H_2O$), methanol (MeOH) and dimethylformamide (DMF) were purchased from FUJIFILM Wako Pure Chemical Corporation Co., Ltd. 3,5-Pyridinedicarboxylic Acid (3,5-pdc) was purchased from Tokyo Chemical Industry Co., Ltd. All chemicals and solvents were used without further purification. Deionized water was used throughout this work. $N_2$, CO, $CO_2$, $C_2H_2$, $O_2$, $H_2$, Ar, $CH_4$, $C_2H_4$, $C_2H_6$ and He were purchased from TAIYO NIPPON SANSO Company (Japan) with a purity of 99.999%.

## Synthesis of PCP-3,5-pdc

First, 194 mg (0.6 mmol) dpg was dissolved in DMF/MeOH (1:1, 60 mL). Then, 150 mg (0.6 mmol) 3,5-pdc and 261 mg (0.6 mmol) $Co(NO_3)_2 \cdot 6H_2O$ were added to the above solution. Then the mixture was heated at 60 °C for 24 h to yield as-synthesised single crystals of PCP-3,5-pdc.

## Activation of PCP-3,5-pdc

To obtain fully desolvated PCP-3,5-pdc, the as-synthesised samples were washed with methanol three times. Finally, the samples were dried under vacuum at 120 °C for 24 h. TGA analysis results indicated that all guest solvents were completely removed (Supplementary Fig. 4).

## General methods

The TGA curves were obtained from a Rigaku TG 8120 analyzer (EVO2 TG/S-SL) using a heating rate of 5 °C min$^{-1}$ in flowing $N_2$. Synchrotron single-crystal X-ray direction (PXRD) patterns were recorded using a RIGAKU RAXIS IV diffractometer (Rigaku, Japan) equipped with a PILATUS3 X CdTe 1M photon counting detector installed in the BL02B1 beamline of SPring-8 ($\lambda = 0.41220$ Å). Single-component $N_2$, CO, $CO_2$, $C_2H_2$, $O_2$, $H_2$, Ar, $CH_4$, $C_2H_4$, $C_2H_6$ sorptions were measured by BEL-mini, BEL-max and BEL-18 (MicrotracBEL Corp., Japan) gas adsorption instruments. The lowest measurement temperature is 60 K for $H_2$ sorption due to the measurement limitation. The pressure of isobar measurement is 100 kPa. Water vapour sorption was measured by Multi-station Gravimetry Vapor Sorption Analyzer 3H-2000PW (BeiShiDe Instrument, Beijing, China).

## High-pressure sorption experiments

High-pressure sorption experiments were carried out by the volumetric BELSORP HP (MicrotracBEL Corp.) instrument (Flow diagram is shown in Supplementary Fig. 12). Prior to the measurement, the blank sample cell weight was measured. The sample was then loaded into the sample cell and heated to 373 K for activation, after which the pre-treated sample was weighted. The adsorption measurement method was the volumetric method, wherein the volume of the measurement system was precisely determined to calculate the volume of adsorption. Then, the volume of adsorption was calculated from the gas pressure change in the measurement system using the gas equation. The dead volume of the sample cell was measured using helium gas of 99.9999% purity. Non-ideal corrections were made by applying virial coefficients at respective measurement temperatures.

## High-pressure co-sorption experiments

High-pressure co-sorption experiments were carried out by the volumetric BELSORP VC (MicrotracBEL Corp.) instrument connected to an Agilent 490 Micro gas chromatographic (GC) system equipped with a thermal conductivity detector (TCD). The flow diagram in Supplementary Fig. 16a[51] provides a visual representation of the experimental setup. In a typical experiment, the sample was first loaded into a pre-weighed stainless-steel sample tube, and activated under dynamic vacuum at 373 K overnight. After activation, the exact sample weight was determined. Then, the sample tube was connected to the instrument and sealed with a metal gasket. Prior to the measurements, the samples were re-heated to 373 K for activation through a removable heater. During measurements, the temperature of the sample was set to 298 K using a removable temperature control unit. Ultra-high purity helium gas (99.9999%) was used to measure the dead volume of the sample tube. The measurement principle is depicted shown in Supplementary Fig. 16b–f. Before the co-adsorption experiment, both gases are stored separately (Supplementary Fig. 16b). To begin the co-adsorption, the gases are introduced into the manifold at the targeted partial pressure and then mixed by the equipped circulation pump (Supplementary Fig. 16c, d). The resulting gas mixture is then introduced into the sample cell (Supplementary Fig. 16e) and adsorbed by the sample (Supplementary Fig. 16f). After a certain equilibration time, the total amount of adsorbed gas is calculated using a constant volume method, and the composition ratio of the adsorbed gas mixture is analysed using GC. The non-adsorbed gas phase over the sample is used to estimate the ratio of the adsorbed gas (Supplementary Fig. 16g). Based on this data, the adsorbed amounts and partial pressures of each gas were calculated.

## Column breakthrough experiments

The breakthrough experiments were carried out using a custom-build dynamic mixed-gas breakthrough setup (Supplementary Fig. 17). In a typical experiment, 1 g of PCP-3,5-pdc sample was packed into a stainless-steel column with inner dimensions of $\phi = 8$ mm. The mixed-gas flow and pressure were controlled by using pressure-control valves (Swagelok) and mass flow controllers (Brooks instrument). Outlet effluent from the column was continuously monitored using a quadrupole-type mass spectrometer, BEL Mass (MicrotracBEL Corp.). The column packed with powder sample was first purged with a flow of He (20 mL·min$^{-1}$) for 1 h at room temperature. The mixed-gas flow rate during the breakthrough process is 6 mL·min$^{-1}$ using 50/50 (v/v) $CO_2$/other gas at room temperatures. The total pressure of the mixture gases was 20 bar. After the breakthrough experiment, the sample was regenerated under vacuum for 24 h for cycling measurement.

## Separation factor calculation

The separation factor is calculated based on the mixture gases' co-adsorption results. The separation factor ($S$) is defined as Eq. (1):

$$S = \frac{X_1/X_2}{Y_1/Y_2} \qquad (1)$$

where $X_1$ and $X_2$ are the concentration of gas 1 and gas 2 in the adsorbed phase and $Y_1$ and $Y_2$ are the concentration of gas 1 and gas 2 in the feed phase.

## Hill analysis

The Hill coefficient, $n$, is recognised as an indicator of cooperative interactions because it describes the number of molecules bound per receptor[52,53]. The Hill coefficient is the slope of the Hill plot {log [$Y/(1 - Y)$] versus log P}, where the $Y$-axis is the $CO_2$ fractional unloading and P is the gas pressure[54,55]. Generally, $n < 1$ corresponds to negatively cooperative systems, while $n > 1$ corresponds to positively cooperative systems. To evaluate the degree of cooperativity for the gate-opening

step in $CO_2$ adsorption, we applied Hill's model analysis to the $CO_2$ adsorption isotherm at 195 and 298 K.

## In-situ PXRD/adsorption measurements

The synchrotron PXRD data for the structural analyses of $CO_2$-loaded PCP-3,5-pdc was collected using a synchrotron X-ray and multiple MYTHEN detectors of the BL02B2 beam line at SPring-8 in Japan[56,57]. The powder sample was put in glass capillaries of 0.5 mm diameter under vacuum. X-ray of wavelength 0.799671(1) Å was selected by a double crystal monochromator using Si(111) plane. The data was taken for 10 min. The lattice parameters and space group were determined by the N-TREOR09 program of EXPO2014 software[58], which was further refined by the Le Bail fit using RIETAN-FP software[59]. A structural starting model for Rietveld refinement was subsequently found with the charge-flipping method using the Superflip and EDMA programs included in RIETAN-FP software based on the extracted lattice parameters and peak intensities from the previous Le Bail fit in space group P−1 (No. 2). The Rietveld refinement was carried out using the RIETAN-FP software using slack soft constraints for bond lengths, angles and planar groups. The final structure of $CO_2$-loaded PCP-3,5-pdc was obtained with reliability factors Rwp = 3.783, Rp = 2.711, S = 4.1301, $R_B$ = 3.480 and $R_F$ = 9.099, respectively. Crystallographic data in CIF format have been deposited in the Cambridge Crystallographic Data Centre (CCDC) under deposition number CCDC-2219713. The data can be obtained free of charge via www.ccdc.cam.ac.uk/data_request/cif (or from the Cambridge Crystallographic Data Centre, 12 Union Road, Cambridge CB2 1EZ, U.K.).

## Single-crystal X-ray crystallography

Suitable crystals of as-synthesised and activated PCP-3,5-pdc were selected for single-crystal X-ray data collection. Intensities were collected on a Rigaku XtaLAB P200 diffractometer using a VariMax Mo Optic with Mo-Kα (λ = 0.71073 Å) equipped with PILATUS 200K detector. These structures were solved by direct methods and refined on $F^2$ by full-matrix least-squares methods with SHELXTL version 2018/3. Some thermal and structural restraints (ISO, REGU, SIMU, SADI, FLAT) and constraint (EADP) on disordered solvent and framework atoms were also used. Hydrogen atoms within the ligand backbones were fixed geometrically and allowed to ride on the parent non-hydrogen atoms in this study. Crystallographic data in CIF format have been deposited in the Cambridge Crystallographic Data Centre (CCDC) under deposition numbers CCDC-2219711 to CCDC-2219712. The data can be obtained free of charge via www.ccdc.cam.ac.uk/data_request/cif (or from the Cambridge Crystallographic Data Centre, 12 Union Road, Cambridge CB2 1EZ, U.K.).

## Computational details

Adsorption energies for several gas species ($CO_2$, $N_2$, $CH_4$, $C_2H_2$, and $C_2H_4$) were carried out to understand the preferential $CO_2$ adsorption over other gas species in PCP-3,5-pdc. Because only $CO_2$ adsorption positions were determined and those of other gas species were unclear in the experiment, we carried out canonical Monte Carlo (MC) simulation[60] to locate their positions in PCP-3,5-pdc in both open and activated phases, as implemented in RASPA[61]. The Lennard–Jones (LJ) potentials were used to describe the Van der Waals interaction of gas molecules with the PCP framework and the electrostatic interaction was evaluated with the Ewald summation method. The LJ parameters for the PCP framework were taken from the standard universal force field (UFF)[62] and the DDEC atomic charges[63,64] of the PCP framework were used in the evaluation of electrostatic interaction. The LJ parameters and atomic charges of $CO_2$, $N_2$, $CH_4$, and $C_2H_4$ were taken from the TraPPE force field[65]. Because there are no reported parameters for $C_2H_2$ in the TraPPE force field, we used those values for the CH moiety in 2-butene[66]. In the MC simulation, the first $1 \times 10^5$ cycles were employed for obtaining equilibration and then $3 \times 10^5$ cycles were used for obtaining a distribution of guest molecules at room temperature.

The final gas adsorption configuration obtained by the above MC simulation was used to construct the initial structure for performing geometry optimisation with density functional theory (DFT).

The binding energy for gas adsorption was calculated using the DFT method with periodic boundary conditions as implemented in the Vienna Ab initio Simulation Package (VASP 5.4.4)[67,68]. The Perdew–Burke–Ernzerhof functional[69] with Grimme's semi-empirical "D3" dispersion term[49] (PBE-D3) was employed in these calculations. The plane wave basis sets with an energy cutoff of 500 eV were used to describe valence electrons and the projector-augmented-wave pseudopotentials[70,71] were used to describe core electrons. The threshold for atomic force convergence was set to be 0.005 eV/Å in geometry optimisation. The Brillouin zone was sampled by a Γ-point in geometry optimisations and 3×3×3 Monkhorst-Pack[72] k-point meshes for energy calculations.

The binding energies (BE) of gas molecules (G, G = $CO_2$, $N_2$, $CH_4$, $C_2H_2$, and $C_2H_4$) with PCP-3,5-pdc were calculated with Eq. (2);

$$BE = E(PCP \cdot nG)_{eq}/n - E(PCP)_{eq}/n - E(G)_{eq} \qquad (2)$$

where $E(PCP \cdot nG)_{eq}$ is the total energy of PCP-3,5-pdc with $n$ gas molecules per unit cell, $E(PCP)_{eq}$ and $E(G)_{eq}$ are the total energies of empty PCP-3,5-pdc and one free gas molecule, respectively, and the subscript "eq" represents the equilibrium structure. The BE was further decomposed into the interaction energies ($E_{int}$) between gas molecules and PCP framework and the deformation energy ($E_{def}$) of the framework induced by gas adsorption. These two energy terms were calculated with Eqs. (3) and (4), respectively;

$$E_{int} = E(PCP \cdot nG)_{eq}/n - E(PCP)_{dis}/n - E(G)_{eq} \qquad (3)$$

$$E_{def} = E(PCP)_{dis} - E(PCP)_{eq} \qquad (4)$$

where $E(PCP)_{dis}$ is the energy of PCP in its distorted structure, which was taken from the equilibrium structure of PCP-3,5-pdc with adsorbed gas molecules. Host-guest and guest-guest interaction energies ($E_{int,H-G}$ and $E_{int,G-G}$) were calculated with Eqs. (5) and (6), respectively;

$$E_{int,H-G} = E(PCP \cdot nG)_{eq}/n - E(PCP)_{dis}/n - E(nG)_{dis}/n \qquad (5)$$

$$E_{int,G-G} = E(nG)_{dis}/n - E(G)_{eq} \qquad (6)$$

where $E(nG)_{dis}$ is the energy of $n$ adsorbed gas molecules taken from their equilibrium adsorption structure.

The climbing-image nudged elastic band (CI-NEB) method[73] was used to evaluate the diffusion barrier of $CO_2$, $C_2H_2$, and $C_2H_4$ in PCP-3,5-pdc. Because the adsorption amount of $C_2H_2$ and $C_2H_4$ is less than 1 molecule per unit cell (0.5 molecule per $Co^{2+}$), we considered only the adsorption with one gas molecule. The convergence criterion for geometry optimisation of the transition state was chosen to be 0.02 eV/Å. Vibrational frequency calculations were carried out to confirm that the initial and transition states have no and one imaginary frequency, respectively.

## Data availability

The crystallographic data for the structures of as-synthesised PCP-3,5-pdc, activated PCP-3,5-pdc and $CO_2$-loaded PCP-3,5-bdc at 195 K in this work have been deposited at the Cambridge Crystallographic Data Centre (CCDC) under deposition numbers CCDC 2219711, 2219713 and 2219714, respectively. These data can be obtained free of charge from the Cambridge Crystallographic Data Centre via https://www.ccdc.cam.ac.uk. The datasets generated during and/or analysed during the current study are available from the corresponding author upon request.

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

## Acknowledgements

This work was supported by the National Key Research and Development Program of China (2022YFE0110500) to F.L. and S.Ki., Shanghai Pujiang Program (NO.21PJ1412600) and the Fundamental Research Funds for the Central Universities of China to Y.G., KAKENHI Grant-in-Aid for Scientific Research (S) (JP18H05262, JP22H05005) and (C) (JP22K05128) from the Japan Society of the Promotion of Science (JSPS). K.O. acknowledges "Yazaki Memorial Foundation for Science and Technology" for the financial support. Synchrotron X-ray measurements were supported by the Japan Synchrotron Radiation Research Institute (JASRI) (Proposal No. 2020A1469, 2020A0617, 2021A1104, 2021A1682). We thank the iCeMS Analysis Center for access to the analytical instruments. We also thank Dr. Mickaele Bonneau for the high-pressure gas sorption measurements.

## Author contributions

Y.G., F.L. and S.Ki. conceived the idea. Y.G., K.O. and S.Ki. designed the experiments. Y.G. and K.O. did the sample preparation, characterisations, gas sorption, and breakthrough measurements. Y.G. and K.O. analysed data. J.Z. and S.S. carried out calculation studies. K.O., H.A., Y.K. and S.Ka. contributed to the synchrotron in-situ PXRD measurements in BL02B2, SPring-8. M.Y. and P.W. helped with the characterisations. Y.G., J.Z., S.S., Y.W., F.L., K.O. and S.Ki. wrote the paper.

## Competing interests

The authors declare no competing interests.
