## [Peer Review File · Nature Communications]

Reviewers' Comments:

Reviewer #1:

Remarks to the Author:

In this paper, authors prepared a novel MOF PCP-3,5-pdc and this MOF exhibited unprecedented selectivity to carbon dioxide over various gas molecules in high pressure through extraordinary gate-opening phenomenon via phase transition of MOF. The unique affinity of CO₂ was confirmed by the mixture gas co-sorption experiment, high-pressure sorption, and high-pressure breakthrough. Furthermore, the underlying mechanism of this phase-transition-based gate-opening behavior of PCP-3,5-Pdc was successfully rationalized by gas-loaded Rietveld refinement, DFT calculation, and energy barrier calculation. And the additional adsorption isotherms which vary the exposure time further underpinned the author's claim that kinetic factor matter when it comes to differentiating similar molecules such as acetylene and ethylene.

I believe that this research has a significant impact because selectively capturing carbon dioxide is crucial for applications including carbon capture, purifying hydrocarbon, and so on. Moreover, this kind of gate-opening due to the guest molecule in the high-pressure region is barely reported. Last but not least, I believe that the gate-opening phenomenon presented in this paper has much potential in terms of practical application compared to gate-opening occurring under 1 bar since conventionally high-pressure gas or compressed gas were used in the practical application field. However, the experimental detail or explanation of high-pressure techniques (co-sorption experiments, high-pressure sorption, and breakthrough experiments) was not sufficient. (especially, in the case of high-pressure isotherm there were any explanations at all except for the acknowledgment section) And given that standardized protocols and procedures for high-pressure experiments were not constructed yet among the researchers and the fact that the result of high-pressure experiments has a high chance of being affected by a systematic error, I believe throughout the explanation of experimental detail of high-pressure experiments were mandatory. To sum up, I would like to recommend this paper to be published after the author reflect on the following recommendation.

1. Author must add the experimental detail of high-pressure co-sorption experiments, high-pressure sorption measurements, and high-pressure breakthrough respectively.
2. The explanation or reference of co-sorption measurements by BEL-VC is required. I don't get how it differentiates the amount of adsorption in the mixed gas state.
3. The explanation and implication of Hill analysis (Figure S7, S11) are needed
4. The additional explanation of separation factor calculation through co-adsorption isotherm (Figure 3d) is required. And comparison with other references is also recommended.
5. The authors state that "Moreover, after the high-pressure adsorption test and natural gas processing (up to 50 bar)". Therefore, it would be better to add the references related to the high-pressure application which the authors mentioned.
6. In the case of a breakthrough, they only conducted CO₂/CH₄ and CO₂/N₂ measurements. As they claimed that PCP-3,5-pdc could separate CO₂ from similar gas molecules, breakthrough experiments much more relevant to there's opinion, such as CO₂/C₂H₂, and CO₂/C₂H₄ breakthrough experiments were required.
7. According to Figures S18, 19, and 20, It seems like the presence of vapor did not affect the structure of PCP-3,5-pdc. However, the effect of humidity in terms of separation performance is not proved. Since structure collapse and the diminution of capturing or separation performance in humid condition were big issue in the CO₂ adsorption field, additional experiments that can prove the effect of humidity is recommended. (For example, humid condition breakthrough or co-sorption between vapor and CO₂ and so on...)

Reviewer #2:

Remarks to the Author:

In "Soft Corrugated Channel with Synergistic Exclusive Discrimination Gating for CO₂ Recognition in Multicomponent Gas Mixture" Gu et al. describe the exclusive discrimination gating (EDG) effect in a porous coordination polymer (PCP) for selective CO₂ adsorption over a wide range of other gases. The work strongly builds on a legacy of similar dynamic effects in porous solids by the group but still provides novel and intriguing insights. I find the manuscript well-structured and illustrated, the experimental details and volume of the data presented and discussed is extensive

and supported by additional computational investigations. Congratulations! While I mostly agree with the interpretation and discussion provided I think that particularly the computational investigation does not live up to the level of the experimental investigations. As such there are a few comments which I would like the authors to address with an otherwise great paper that should be published in Nature Comms.

The single component adsorption isotherms were recorded at the standard boiling points of the respective gases. Many of the conditions are quite unusual for adsorption investigations. I recognize that similar temperature ranges have been investigated by Kaskel and co-workers in this paper: <https://pubs.rsc.org/en/content/articlelanding/2021/fd/d0fd00013b> Is this temperature selected to record the full relative pressure range up to 100 kPa or is the reasoning different? Can the authors detail to why these conditions were chosen and how they might influence the described selectivity?

A long standing question in the field of soft porous crystals with respect to selective adsorption is the co-adsorption of various guest species after pore-opening has occurred. The authors describe the size-selective adsorption sites for CO₂ and hydrocarbons but did not discuss the (quasi)equilibrium situation of co-adsorption after pore opening in details. There is only a very short section that details experiments which are otherwise "hidden" in the ESI. I suggest to comment on this aspect a bit more in detail. For example if the CO₂ is blocking the pore entry and prevents effective exchange in the pore network etc.

In the main text and ESI the phrase "exposure time" correlates to the equilibration time of individual adsorption steps? A clarification might be helpful given that the authors not only record isotherms but also isobars.

I suggest to add a schematic energy diagram in Figure 5 (similar to figure 4) that supports and summarizes the evolution of the energy landscape in this system.

In the computational analysis the authors describe by DFT the evolution of binding energies vs loading of various adsorption sites. Did the authors consider the contribution of CO₂-CO₂ (or more general gas-gas) interactions in these calculations and can they derive to what extend these interactions contribute the less favorable adsorption energetics with enhanced loading? The authors state: "These results indicate that the selective adsorption of CO₂ over C₂H₂ and C₂H₄ was not thermodynamically but kinetically controlled." In the simulations only single gas scenarios have been computed and although I agree that the data of this work indicates this scenario the computation is the weakest link in this regard. Maybe it would be helpful to refer to the experimental data or draw a detailed comparison.

The authors use the climbing-image nudged elastic band (CI-NEB) method to evaluate the diffusion barrier. However, in this model the framework is treated as a rigid entity which based on the experimental finding is certainly not the case in this system. In addition, this approach provides no insight into the co-diffusion in gas mixtures in which CO₂ may allow to open the pore structure allowing other gases to enter as well. I am thus skeptical to what degree these calculations are an adequate description of the real world scenario and in fact to what degree they can contribute to establish the underlying mechanism. Did the authors try to apply molecular dynamics simulations in this system as this is an established method to reliably determine diffusion processes as well as structural deformations in PCPs and other dynamic porous solids.

Reviewer #3:

Remarks to the Author:

Prof. Kitagawa and coworkers reported a flexible PCP showing highly selective CO₂ adsorption over 9 typical gases, which has not been achieved by other materials. While most references still focus on separation of simple/ideal mixtures containing two or three components, this result would call attention for separation of highly complicated mixtures. I strongly suggest publication of this work with some minor concerns.

1. A few references have reported highly selective adsorption of CO₂ over hydrocarbons or over inorganic gases. It is better to compare the results with the known examples, including performances and mechanisms.
2. Fig. 1d is claimed as synergistically utilizing all available recognition mechanisms as shown in Fig. 1a-c. However, the concept in Fig.1a is not involved in Fig. 1d. For the computational simulation of the diffusion barrier, the dynamics or the transient structural distortion of the narrow window is considered, which is consistent with the concept shown in Fig. 1d. Therefore, it is suggested to revise Fig. 1a according to that of Fig.1d and the computational simulations.
3. Are the two large circular pores in Fig. 1d represent rigid or soft? If rigid, it is not the case for the titled PCP. If soft, the structural transformation is not shown.
4. The term Exclusive Discrimination seems to refer to an ideal selectivity. However, as the manuscript stated, many of the 9 gases can be adsorbed. In the literature, thermodynamic separation would not be stated as exclusive/ideal even when the selectivity is extremely high. It would be better to revise or clarify the term.
5. The orientation of Fig 2a is not suitable to reveal the structural relationship with that in Fig. 2b. Please describe how many Co(II) ions are coordinated to the 3,5-pdc²⁻ and dpg ligands, in all three phases. It is suggested to draw the simplified topological structures of the three phases.
6. As shown in Fig. 2d-e, the uptake below the gate-opening pressure is attributed to the inclusion of CO₂ in the intrinsic microporous cavities of phase beta. However, phase beta is claimed as dense in other places. Please clearly compare the crystallographic pore parameters (in a table) and the pore structures (in a figure) of the three phases. Before doing these, please polish the crystal structures to eliminate the errors such as A/B-level alerts, missing hydrogen atoms, nonplanar aromatic rings, etc.
7. About the Hill coefficient, reference should be cited, and the discussion seems to be too simple and meaningless.
8. ...CO₂ required activation energy to adsorb onto... What's the meaning of activation energy? ...such adsorption did not occur... Which is "such adsorption"?
9. Specify the pressure of isobars. Specify the phase used for computational simulations.
10. The volume of N₂, CH₄, and C₂H₄ adsorbed in the binary adsorption mixtures was negligible. They should not be claimed as negligible. If negligible, the uptakes should not be compared and calculated. ... optimum selectivity... How optimum? This is a typical feature of gate-opening adsorption. What is "this"?

Response to the reviewer comments

Journal Name: Nature communications

Manuscript Title: Soft Corrugated Channel with Synergistic Exclusive Discrimination Gating for CO₂ Recognition in Multicomponent Gas Mixture

Manuscript No: NCOMMS-23-07080-T

Thank the referees for their helpful comments and suggestions. Based on these comments and suggestions, related changes have been done in the revised manuscript as follows:

Reviewer #1: In this paper, authors prepared a novel MOF PCP-3,5-pdc and this MOF exhibited unprecedented selectivity to carbon dioxide over various gas molecules in high pressure through extraordinary gate-opening phenomenon via phase transition of MOF. The unique affinity of CO₂ was confirmed by the mixture gas co-sorption experiment, high-pressure sorption, and high-pressure breakthrough. Furthermore, the underlying mechanism of this phase-transition-based gate-opening behavior of PCP-3,5-pdc was successfully rationalized by gas-loaded Rietveld refinement, DFT calculation, and energy barrier calculation. And the additional adsorption isotherms which vary the exposure time further underpinned the author's claim that kinetic factor matter when it comes to differentiating similar molecules such as acetylene and ethylene. I believe that this research has a significant impact because selectively capturing carbon dioxide is crucial for applications including carbon capture, purifying hydrocarbon, and so on. Moreover, this kind of gate-opening due to the guest molecule in the high-pressure region is barely reported Last but not least, I believe that the gate-opening phenomenon presented in this paper has much potential in terms of practical application compared to gate-opening occurring under 1 bar since conventionally high-pressure gas or compressed gas were used in the practical application field.

However, the experimental detail or explanation of high-pressure techniques (co-sorption experiments, high-pressure sorption, and breakthrough experiments) was not sufficient. (Especially, in the case of high-pressure isotherm there were any explanations at all except for the acknowledgment section) And given that standardized protocols and procedures for high-pressure experiments were not constructed yet among the researchers and the fact that the result of high-pressure experiments has a high chance of being affected by a systematic error, I believe throughout the explanation of experimental detail of high-pressure experiments were mandatory. To sum up, I would like to recommend this paper to be published after the author reflect on the following recommendation.

Response

We are grateful to the reviewer for their time and effort in evaluating our manuscript and recognizing its “significant impact”. We really appreciate the reviewer’s constructive comments to improve the quality of our manuscript. We have carefully considered the remarks mentioned by the reviewer and have made the relevant revisions to the manuscript. Notably, we have added an explanation of the high-pressure

experiments.

1. Author must add the experimental detail of high-pressure co-sorption experiments, high-pressure sorption measurements, and high-pressure breakthrough respectively.

Response

We thank the reviewer for this comment. We have included the experimental detail of high-pressure co-sorption experiments, high-pressure sorption measurements, and high-pressure breakthrough experiments. The relevant information is outlined below.

High-pressure Sorption Experiments (Page S3 and S22)

High-pressure sorption experiments were carried out by the volumetric BELSORP HP (MicrotracBEL Corp.) instrument (Flow diagram is shown in Supplementary Figure 12). Prior to the measurement, the blank sample cell weight was measured. The sample was then loaded into the sample cell and heated to 373 K for activation, after which the pretreated sample was weighted. The adsorption measurement method was the volumetric method, wherein the volume of measurement system was precisely determined to calculate the volume of adsorption. Then, the volume of adsorption was calculated from the gas pressure change in the measurement system using the gas equation. The dead volume of the sample cell was measured using helium gas of 99.9999% purity. Non-ideal corrections were made by applying virial coefficients at respective measurement temperatures.

Supplementary Figure 12. Measurement set-up of the used BELSORP HP instrument for high-pressure sorption experiments (AV = air-operated valve, NV = needle valve, CV = check valve, P / PS= pressure sensor). The flow diagram is shown in green color.

High-pressure Co-sorption Experiments (Page S3 and S26)

High-pressure co-sorption experiments were carried out by the volumetric BELSORP VC (MicrotracBEL Corp.) instrument connected to an Agilent 490 Micro gas chromatographic (GC) system equipped with a thermal conductivity detector (TCD). The flow diagram in Supplementary Figure 16a¹ provides a visual representation of the experimental setup. In a typical experiment, the sample was first loaded into a pre-weighed stainless-steel sample tube, and activated under dynamic vacuum at 373 K overnight. After activation, the exact sample weight was determined. Then, the sample tube was connected with the instrument and sealed with a metal gasket. Prior to the measurements, the samples were re-heated to 373 K for activation through removeable heater. During measurements, the temperature of the sample was set to be 298 K using a removable temperature control unit. Ultra-high purity helium gas (99.9999%) was used to measure the dead volume of the sample tube. The measurement principle is depicted shown in Supplementary Figure 16b-f. Before the co-adsorption experiment, both gases are stored separately (Supplementary Figure 16b). To begin the co-adsorption, the gases are introduced into the manifold at the targeted partial pressure and then mixed by the equipped circulation pump (Supplementary Figure 16c and 16d). The resulting gas mixture is then introduced into the sample cell (Supplementary Figure 16e) and adsorbed by the sample (Supplementary Figure 16f). After a certain equilibration time, the total amount of adsorbed gas is calculated using a constant volume method, and the composition ratio of the adsorbed gas mixture is analyzed using GC. The non-adsorbed gas phase over the sample is used to estimate the ratio of the adsorbed gas (Supplementary Figure 16g). Based on this data, the adsorbed amounts and partial pressures of each gas were calculated.

Supplementary Figure 16. (a) Measurement set-up of the used Belsorp VC instrument for high-pressure co-sorption experiments. (b-g) Description of the measurement principle step-by-step¹.

Breakthrough Experiments (Page S4 and S27)

The breakthrough experiments were carried out using a custom-built dynamic mixed-gas breakthrough setup (Supplementary Figure 17). In a typical experiment, 1 g of PCP-3,5-pdc sample was packed into a stainless-steel column with inner dimensions of $\phi = 8$ mm. The mixed-gas flow and pressure were controlled by using pressure-

control valves (Swagelok) and mass flow controllers (Brooks instrument). Outlet effluent from the column was continuously monitored using a quadrupole-type mass spectrometer, BEL Mass (MicrotracBEL Corp.). The column packed with powder sample was first purged with a flow of He ($20 \text{ mL} \cdot \text{min}^{-1}$) for 1 h at room temperature. The mixed-gas flow rate during the breakthrough process is $6 \text{ mL} \cdot \text{min}^{-1}$ using 50/50 (v/v) CO_2 /other gas at room temperatures. The total pressure of mixture gases was 20 bar. After the breakthrough experiment, the sample was regenerated under vacuum 24 hours for cycling measurement.

Supplementary Figure 17. The high-pressure breakthrough system used in this study.

2. The explanation or reference of co-sorption measurements by BEL-VC is required. I don't get how it differentiates the amount of adsorption in the mixed gas state.

Response

We thank the reviewer for this comment. The detailed explanation of the co-sorption measurement can be found in our response above.

3. The explanation and implication of Hill analysis (Figure S7, S11) are needed.

Response

We thank the reviewer for this comment. We have added an explanation and implication of Hill analysis in the Supplementary Information (Page S4) as follows.

The Hill coefficient, n , is recognized as an indicator of cooperative interactions because it describes the number of molecules bound per receptor^{2,3}. The Hill coefficient is the slope of the Hill plot $\{\log [Y/(1-Y)] \text{ versus } \log P\}$, where Y-axis is the CO_2 fractional unloading and P is the gas pressure^{4,5}. Generally, $n < 1$ corresponds to negatively cooperative systems, while $n > 1$ corresponds to positively cooperative systems. To evaluate the degree of cooperativity for the gate-opening step in CO_2 adsorption, we applied Hill's model analysis to the CO_2 adsorption isotherm at 195 K and 298 K.

The Hill coefficient for the gate-opening step in the CO₂ adsorption isotherm at 195 K was determined to be 4.1 (>1) (Supplementary Figure 9), confirming the positive, cooperative adsorption phenomenon. The positive CO₂ adsorption cooperativity in the structural transformation step was even stronger (Hill coefficient $n = 6.6$, Supplementary Figure 14). The difference in cooperativity at different temperature may be due to the varying diffusion and stabilizing abilities of the CO₂ molecules.

4. The additional explanation of separation factor calculation through co-adsorption isotherm (Figure 3d) is required. And comparison with other references is also recommended.

Response

We thank the reviewer for this comment. The separation factor is calculated based on the mixture gases co-adsorption results. The separation factor (S) is defined as:

$$S = \frac{X_1 / X_2}{Y_1 / Y_2}$$

where X_1 and X_2 are the concentration of gas 1 and 2 in the adsorbed phase and Y_1 and Y_2 are the concentration of gas 1 and 2 in the feed phase. This information is provided in the Supplementary Information (Page S4).

We also thank the reviewers for suggesting comparison with other references. Usually, IAST selectivity, which is based on the ideal adsorbed solution theory, is commonly used to evaluate the potential of MOFs for capturing CO₂ from gas mixtures. However, this method relies on adsorption in a solution phase, or at temperature near the boiling point of the gases being adsorbed, and is usually calculated from a single-component adsorption isotherm. As a result, it may not fully represent MOF selectivity in gas separation scenarios, especially those involving structural flexibility^{6, 7}. In this study, we have calculated the separation factor based on the results of co-adsorption experiments with gas mixtures. This approach differs from the commonly used IAST selectivity index and is difficult to compare directly. To contextualize our results, we have compared them with other references that primarily focused on general CO₂ capture performances and separation mechanisms (Supplementary Table 8, page S53).

Supplementary Table 8. The CO₂ selective adsorption performance and main recognition mechanisms in representative PCPs

PCP	Reported CO ₂ selective adsorption performance	Main mechanisms
Qc-5-Cu-sql ⁸	Selective CO ₂ adsorption over CH ₄ and N ₂	
Zr-FA ⁹	Selective CO ₂ adsorption over CH ₄ and N ₂	Molecular sieving
[Cu(bcppm)H ₂ O] ¹⁰	Selective CO ₂ adsorption over N ₂	
M-gallate (M = Mg, Co, Ni) ¹¹	Selective CO ₂ adsorption over CH ₄ and N ₂	

ZU-610a ¹²	Selective CO ₂ adsorption over C ₂ H ₂	Kinetic sieving
[Cu ₂ (ndpa)] ¹³	Selective CO ₂ adsorption over CH ₄ and N ₂	Interactions with open metal site
CPO-27-M (M = Mg, Co, Ni) ¹⁴	Selective CO ₂ adsorption over CH ₄ and N ₂	
[Cu ₂ (OH) ₂ (bdim)] ¹⁵	Selective CO ₂ adsorption over N ₂	Pore size and electrostatic potential complementary
MUF-16-M (M = Co, Mn, Ni) ¹⁶	Selective CO ₂ adsorption over CH ₄ , C ₂ H ₂ , C ₂ H ₄ , C ₂ H ₆ , C ₃ H ₆ and C ₃ H ₈	
[Cu(tba) ₂] ¹⁷	Selective CO ₂ adsorption over CH ₄ , N ₂ , O ₂ , Ar, and H ₂	host-guest C-H...O and guest-guest interactions
[Mg ₂ (dobpdc)(eda) _{1.6}] ¹⁸	Direct air capture and CO ₂ capture from flue gas	Interactions with appended amine
[Mg ₂ (dobpdc)(mmen) _{1.6}] ¹⁹	Direct air capture and CO ₂ capture from flue gas	
Mg ₂ (dobpdc)(N,N'-bis(3-aminopropyl)-1,4-diaminobutane) ²⁰	Direct air capture and CO ₂ capture from flue gas	
SIFSIX-3-Cu ²¹	Direct air capture	Multiple weak interactions
SIFSIX-3-Zn ²²	Selective CO ₂ adsorption over CH ₄ , N ₂ , and H ₂	Multiple weak interactions
MAF-66 ²³	Selective CO ₂ adsorption over CH ₄ and N ₂	Interactions with multiple N
Co(bdp) ²⁴	Selective CO ₂ adsorption over CH ₄	The cooperation of framework flexibility and molecular sieving
[Zn(5NO ₂ -ip)(dpe)] ²⁵	Selective CO ₂ adsorption over CH ₄ , C ₂ H ₄ and C ₂ H ₆	Framework flexibility
[Mn(bdc)(dpe)] ²⁶	Selective CO ₂ adsorption over C ₂ H ₂	Framework flexibility
PCP-3,5-pdc (this work)	Selective CO ₂ adsorption over N ₂ , CH ₄ , CO, O ₂ , H ₂ , Ar, C ₂ H ₂ , C ₂ H ₄ and C ₂ H ₆	The cooperation of pore stereochemical shape, location of binding sites, and framework flexibility

5. The authors state that “Moreover, after the high-pressure adsorption test and natural gas processing (up to 50 bar)”. Therefore, it would be better to add the references related to the high-pressure application which the authors mentioned.

Response

We thank the reviewer for this comment. As per the reviewer’s advice, we have included the following references relevant to the high-pressure application:

- Hu et al provided the pressure range (20-30 bar) for precombustion CO₂ capture in their review work published in *Advanced Sustainable Systems* in 2019 (DOI:10.1002/adsu.201800080).

• Qazvini et al conducted the breakthrough simulations at pressures relevant to natural gas processing (50 bar) and predict that MUF-16 could capture CO₂ from natural gas. This work was reported in Nature Communications in 2021 (DOI: 10.1038/s41467-020-20489-2).

We have added these references to the revised manuscript (page 11).

6. In the case of a breakthrough, they only conducted CO₂/CH₄ and CO₂/N₂ measurements. As they claimed that PCP-3,5-pdc could separate CO₂ from similar gas molecules, breakthrough experiments much more relevant to there's opinion, such as CO₂/C₂H₂, and CO₂/C₂H₄ breakthrough experiments were required.

Response

We appreciate the reviewer's comment. Conducting the breakthrough experiments with C₂H₂ is not feasible because we need to perform the experiment under high pressure, and it would reach the explosive limit of C₂H₂ gas. In response to the reviewer's suggestion, we further performed CO₂/C₂H₄ breakthrough experiments as shown in Supplementary Figure 21 (page 32 in the Supplementary Information), which also support the CO₂ preference over C₂H₄ under the given condition.

Supplementary Figure 21. Experimental breakthrough curve of PCP-3,5-pdc at a flow rate of 6 mL/min for an equimolar gaseous mixture of C₂H₄ and CO₂ (v/v, 50/50) at room temperature (Total pressure 20 bar).

7. According to Figures S18, 19, and 20, It seems like the presence of vapor did not affect the structure of PCP-3,5-pdc. However, the effect of humidity in terms of separation performance is not proved. Since structure collapse and the diminution of

capturing or separation performance in humid condition were big issue in the CO₂ adsorption field, additional experiments that can prove the effect of humidity is recommended. (For example, humid condition breakthrough or co-sorption between vapor and CO₂ and so on...)

Response

We are grateful to the reviewer for providing constructive feedback. At high-pressure conditions, the condensation of water vapor may occur. Therefore, due to the high-pressure conditions required for our breakthrough or co-sorption experiments, it was not possible to perform the experiment in a humid environment. Instead, we investigated the impact of humidity on the separation performance by comparing the CO₂/N₂ breakthrough curves of the sample exposed to humid air for more than one week, both before and after thermal activation. Supplementary Figure S25 (Page S35) illustrates the CO₂/N₂ breakthrough curves of the sample, before and after activation (at 373 K under vacuum for 2 hours). The results indicate that while the CO₂/N₂ separation ability was sustained, its performance degraded due to humidity. However, since the structure remained stable even under humid conditions, the separation performance was not affected after activation (page 12 in the revised manuscript).

Supplementary Figure 25. Experimental breakthrough curve of PCP-3,5-pdc before and after activation (at 373 K under vacuum for 2 hours), after exposure to humidity for more than one week at a flow rate of 6 mL/min for an equimolar gaseous mixture of N₂ and CO₂ (v/v, 50/50) at room temperature (Total pressure 20 bar).

Reviewer #2: In “Soft Corrugated Channel with Synergistic Exclusive Discrimination Gating for CO₂ Recognition in Multicomponent Gas Mixture” Gu et al. describe the exclusive discrimination gating (EDG) effect in a porous coordination polymer (PCP) for selective CO₂ adsorption over a wide range of other gases. The work strongly builds on a legacy of similar dynamic effects in porous solids by the group but still provides novel and intriguing insights. I find the manuscript well-structured and illustrated, the experimental details and volume of the data presented and discussed is extensive and supported by additional computational investigations. Congratulations! While I mostly agree with the interpretation and discussion provided. I think that particularly the computational investigation does not live up to the level of the experimental investigations. As such there are a few comments which I would like the authors to address with an otherwise great paper that should be published in Nature Comms.

Response

We thank the reviewer for reviewing our manuscript and recognizing it as “a great paper”. We really appreciate the constructive comments that helped to improve the quality of our manuscript. We carefully considered all the remarks mentioned and made related revisions to the manuscript.

1. The single component adsorption isotherms were recorded at the standard boiling points of the respective gases. Many of the conditions are quite unusual for adsorption investigations. I recognize that similar temperature ranges have been investigated by Kaskel and co-workers in this paper: <https://pubs.rsc.org/en/content/articlelanding/2021/fd/d0fd00013b>. Is this temperature selected to record the full relative pressure range up to 100 kPa or is the reasoning different? Can the authors detail to why these conditions were chosen and how they might influence the described selectivity?

Response

We thank the reviewer for this comment. In this work, we first measured the single component adsorption isotherms at their respective standard boiling points, up to 100 kPa (Fig. 2d, page 7 in the revised manuscript). A lower temperature provides stronger host-guest interactions from thermodynamic perspective. Therefore, the single component adsorption isotherms at the standard boiling points of each gas were recorded over the pressure range up to 100 kPa to examine the gas-dependent sorption behavior of PCP-3,5-pdc. In addition, we also recorded the single-component adsorption isotherms for each gas at the same temperature of 298 K, up to 30 bar (Supplementary Figure 13, page S23). These results confirmed that, except CO₂, none of the other gases induced the gate-opening behavior of PCP-3,5-pdc. This unique sorption behavior is unprecedented and therefore, interesting. This unique sorption behavior is unprecedented and, therefore, we believe it is interesting.

2. A long-standing question in the field of soft porous crystals with respect to selective adsorption is the co-adsorption of various guest species after pore-opening has occurred.

The authors describe the size-selective adsorption sites for CO₂ and hydrocarbons but did not discuss the (quasi)equilibrium situation of co-adsorption after pore opening in details. There is only a very short section that details experiment which are otherwise “hidden” in the ESI. I suggest to comment on this aspect a bit more in detail. For example, if the CO₂ is blocking the pore entry and prevents effective exchange in the pore network etc.

Response

We thank the reviewer for this comment. The co-adsorption of various guest species in soft porous crystals depends on their affinities to the framework, even after pore-opening has occurred. In the case of co-adsorption of CO₂ and C₂H₄ in PCP-3,5-pdc, C₂H₄ is more adsorbed at room temperature than CO₂ before the opening of pores. As pressure increases, the pores in this PCP can be opened by CO₂ adsorption. However, the adsorption of C₂H₄ did not increase compared to that before the pore opening, suggesting that guest exchange is difficult to occur (Fig. 3c, page 10 in the revised manuscript). These co-adsorption results suggest that PCP-3,5-pdc has a stronger affinity for C₂H₄ than for CO₂ before the pore-opening occurs, but a weaker affinity for C₂H₄ than for CO₂ after the pore-opening has occurred. This is consistent with the calculated binding energies of these gas molecules with this PCP at different pore-opening states. In this regard, we agree with the reviewer’s suggestion to discuss further and have added a sentence in page 11 of the revised manuscript: “These results suggest that the adsorption of CO₂ in the open framework of PCP-3,5-pdc is strong enough to block the pore entry, preventing effective guest exchange in the pore network.”.

3. In the main text and ESI the phrase “exposure time” correlates to the equilibration time of individual adsorption steps? A clarification might be helpful given that the authors not only record isotherms but also isobars.

Response

We thank the reviewer for this comment. The exposure time is correlates to the equilibration time. We have used exposure time throughout the main text and Supplementary Information in this revised version.

4. I suggest to add a schematic energy diagram in Figure 5 (similar to figure 4) that supports and summarizes the evolution of the energy landscape in this system.

Response

We thank the reviewer for this suggestion. We have added a schematic energy diagram in the revised Figure 5. The revised Figure 5 is shown as follows (Page 16 in the revised manuscript).

Fig. 5 | A schematic diagram of the diffusion of CO₂, C₂H₂ and C₂H₄ in the narrow-corrugated channel of PCP-3,5-pdc. a, CO₂ molecule at initial state (IS), transition state (TS) for diffusion of CO₂ through the narrow window, and the diffusion pathway of CO₂ in the 1-D corrugated channel. b, C₂H₄ molecule at initial state (IS), transition state (TS) for diffusion of CO₂ through the narrow window, and the diffusion pathway of CO₂ in the 1-D corrugated channel. c, C₂H₂ molecule at initial state (IS), transition state (TS) for diffusion of CO₂ through the narrow window, and the diffusion pathway of CO₂ in the 1-D corrugated channel. Purple, Red, blue, grey and pink in the PCP frameworks and gas molecules represent Co, O, N, C and H, respectively. The diffusion barrier was calculated using the computational model with one adsorbed gas molecule. We considered a diffusion process in which gas molecules moves from site I to its neighbouring site II. Interaction energies (E_{int}), deformation energies (E_{def}), and diffusion barriers (E_b) are given in kcal mol⁻¹.

5. In the computational analysis the authors describe by DFT the evolution of binding energies vs loading of various adsorption sites. Did the authors consider the contribution of CO₂-CO₂ (or more general gas-gas) interactions in these calculations and can they derive to what extent these interactions contribute the less favorable adsorption energetics with enhanced loading?

Response

We thank the reviewer for this comment. Gas-gas interactions were considered in our DFT calculations. As shown in Supplementary Table 7 (Page S44), the gas-gas

interactions ($E_{\text{int, G-G}}$) becomes more negative with enhanced CO₂ loading, which is favorable for CO₂ adsorption. However, as the number of CO₂ molecules per unit cell increases to 4, the $E_{\text{int, H-G}}$ becomes less negative, which is unfavorable for further CO₂ adsorption. We revised the corresponding discussion in page 15 as follows:

The increase in BE arises from the formation of T-shaped molecular clusters of CO₂ between the adsorbed CO₂ molecules stabilizing gas-gas interactions, which stabilizes gas-gas interactions, and the decrease in average deformation energy of PCP induced by CO₂ adsorptions (Fig. 4a and Supplementary Table 7). However, the BE value decreases (less negative) when four CO₂ molecules are adsorbed at all the adsorption sites II to afford PCP·4CO₂ (Fig. 4c) because of the congestion due to the ‘narrow-corrugated channel’, as suggested by the decreased interaction energy ($E_{\text{int, H-G}}$) between CO₂ molecules and PCP framework (Supplementary Table 7).

6. The authors’ state: “These results indicate that the selective adsorption of CO₂ over C₂H₂ and C₂H₄ was not thermodynamically but kinetically controlled.” In the simulations only single gas scenarios have been computed and although I agree that the data of this work indicates this scenario the computation is the weakest link in this regard. Maybe it would be helpful to refer to the experimental data or draw a detailed comparison.

Response

We thank the reviewer for this comment and agree with this reviewer that our original statement might be too strong without referring to experimental data. According to the calculated binding energy, PCP-3,5-pdc has a stronger affinity to C₂H₂ or C₂H₄ than to CO₂. Therefore, this PCP can should selectively adsorb C₂H₂ or C₂H₄ over CO₂, which is not observed at low temperature. Actually, experimental results of high-pressure gas adsorption at room temperature also support that PCP-3,5-pdc has a stronger affinity to C₂H₂ or C₂H₄ than to CO₂. Therefore, we inferred that kinetic factors may play important roles for the selective adsorption of CO₂ over C₂H₂ and C₂H₄, which were discussed in the subsequent section by referring to both computational and experimental data. Therefore, we revised the sentence as “These results suggest that the selective adsorption of CO₂ over C₂H₂ and C₂H₄ at low temperature was not thermodynamically but could be kinetically controlled, as discussed below.” for clarity (Page 17 in the revised manuscript).

7. The authors use the climbing-image nudged elastic band (CI-NEB) method to evaluate the diffusion barrier. However, in this model the framework is treated as a rigid entity which based on the experimental finding is certainly not the case in this system. In addition, this approach provides no insight into the co-diffusion in gas mixtures in which CO₂ may allow to open the pore structure allowing other gases to enter as well. I am thus skeptical to what degree these calculations are an adequate description of the real world scenario and in fact to what degree they can contribute to establish the underlying mechanism. Did the authors try to apply molecular dynamics simulations in

this system as this is an established method to reliably determine diffusion processes as well as structural deformations in PCPs and other dynamic porous solids.

Response

We thank the reviewer for this comment. As pointed out by this reviewer, co-adsorption and co-diffusion of gas mixtures are open questions in the field of soft porous crystals. It is still unclear whether the pore opened by adsorption of one gas molecule can keep open permanently and allow adsorption/diffusion of other gas species. To fully address this question, a comprehensive computational study on the equilibrium and non-equilibrium adsorptions using various computational methods is required, as recommended by this reviewer. However, to our understanding, this is a bit beyond the scope of this work. In the present work, gas molecules are adsorbed in the one-dimensional narrow-corrugated channels of PCP-3,5-pdc, which only allows gas adsorption/diffusion in a sequential manner. In addition, the co-adsorption experiment suggest that the pores opened by CO₂ adsorption rarely allow the entry of other gas species. Besides, both room-temperature gas adsorption measurement and DFT calculations on equilibrium gas adsorption suggest that PCP-3,5-pdc should have a stronger affinity to C₂H₄ than to CO₂ before pore-opening has occurred. However, at low temperatures near to the boiling points of gas molecules, the adsorption amount of CO₂ is larger than that of C₂H₄ even before pore-opening has occurred. Therefore, we inferred that calculating the diffusion barriers of various gas molecules could give insight into their different adsorption behaviors. In this regard, we think the CI-NEB method is useful to calculate the diffusion barriers of gas molecules before pore-opening has occurred, which leads to reasonable values to understand the difference in adsorption behavior between CO₂ and hydrocarbons. Given above considerations, we did not try to apply molecular dynamics simulations in this work but will keep in mind while trying to theoretically address the open question of co-adsorption and co-diffusion of gas mixtures in selected soft porous crystals.

Reviewer #3: Prof. Kitagawa and coworkers reported a flexible PCP showing highly selective CO₂ adsorption over 9 typical gases, which has not been achieved by other materials. While most references still focus on separation of simple/ideal mixtures containing two or three components, this result would call attention for separation of highly complicated mixtures. I strongly suggest publication of this work with some minor concerns.

Response

Thanks to the reviewer for reviewing our manuscript and strongly recommending the publication of this work. We really appreciate the constructive comments that helped to improve the quality of our manuscript. We carefully considered all the remarks mentioned and made related revisions to the manuscript.

1. A few references have reported highly selective adsorption of CO₂ over hydrocarbons or over inorganic gases. It is better to compare the results with the known examples, including performances and mechanisms.

Response

We thank the reviewer for this comment. Herein, we added a comparison with other reference mainly focused on general CO₂ capture performances and separation mechanisms (Supplementary Table 8, page S53). This table can be found in our response to Reviewer #1's comment 4.

2. Fig. 1d is claimed as synergistically utilizing all available recognition mechanisms as shown in Fig. 1a-c. However, the concept in Fig. 1a is not involved in Fig. 1d. For the computational simulation of the diffusion barrier, the dynamism or the transient structural distortion of the narrow window is considered, which is consistent with the concept shown in Fig. 1d. Therefore, it is suggested to revise Fig. 1a according to the that of Fig. 1d and the computational simulations.

Response

We thank the reviewer for this comment. We revised the Fig. 1a according to the reviewer's suggestions. The revised Fig. 1a shows the molecular sieving and diffusion regulation mechanisms in rigid PCPs. In the Fig. 1d, the bottleneck aperture in the corrugated channel after gate-opening also can show molecular sieving and diffusion regulation functions. The revised Fig. 1a is shown as follows (Page 5 in the revised manuscript).

3. Are the two large circular pores in Fig. 1d represent rigid or soft? If rigid, it is not the case for the titled PCP. If soft, the structural transformation is not shown.

Response

We thank the reviewer for this comment. We have included the structural transformation in the revised Fig. 1d, which is displayed below.

4. The term Exclusive Discrimination seems to refer to an ideal selectivity. However, as the manuscript stated, many of the 9 gases can be adsorbed. In the literature, thermodynamic separation would not be stated as exclusive/ideal even when the selectivity is extremely high. It would be better to revise or clarify the term.

Response

We thank the reviewer for this comment. In this revised manuscript, we use the term exclusive discrimination gating effect to refer to the selective gate-opening adsorption behavior for CO₂ over other 9 gases. We clarify the term in the revised manuscript (page 2 in the revised manuscript).

5. The orientation of Fig 2a is not suitable to reveal the structural relationship with that in Fig. 2b. Please describe how many Co(II) ions are coordinated to the 3,5-pdc- and dpg ligands, in all three phases. It is suggested to draw the simplified topological structures of the three phases.

Response

We thank the reviewer for this comment. The following Supplementary Figure 3a-c show the coordination environment of Co(II) in as-synthesised, activated and CO₂

loaded PCP-3,5-pdc. In the as-synthesised PCP-3,5-pdc, each Co(II) is coordinated to three 3,5-pdc and three dpg ligands. In the activated and CO₂ loaded PCP-3,5-pdc, each Co(II) is coordinated to three 3,5-pdc and two dpg ligands. As per the reviewer's advice, the simplified topological structures of the three phases have been added to page S12 in the revised Supplementary Information (Supplementary Figure 3d-f).

Supplementary Figure 3. (a-c) The coordination environment of Co(II) in phase α , β and γ of PCP-3,5-pdc. (d-f) The simplified topological structures of phase α , β and γ of PCP-3,5-pdc. In the as-synthesised PCP-3,5-pdc, each Co(II) is coordinated to three 3,5-pdc and three dpg ligands. In the activated and CO₂ loaded PCP-3,5-pdc, each Co(II) is coordinated to three 3,5-pdc and two dpg ligands.

6. As shown in Fig. 2d-e, the uptake below the gate-opening pressure is attributed to the inclusion of CO₂ in the intrinsic microporous cavities of phase beta. However, phase beta is claimed as dense in other places. Please clearly compare the crystallographic pore parameters (in a table) and the pore structures (in a figure) of the three phases. Before doing these, please polish the crystal structures to eliminate the errors such as A/B-level alerts, missing hydrogen atoms, nonplanar aromatic rings, etc.

Response

We thank the reviewer for this comment. As per the reviewer suggestion, we polished the crystal structures while taking care of the A/B-level alerts (the updated results are summarized in Supplementary Table S2, and S3). Regarding the Rietveld analysis result of the CO₂ adsorbed phase (γ), the nonplanar aromatic rings and the missing hydrogen atoms resulted because this structural analysis was performed based on the PXRD pattern analysis without using a rigid-body model (the refinement detail is given in Supplementary Information Page S5, and Supplementary Table S5). Therefore, prior to the structural comparisons of these structure, we computationally optimized the γ

structure by adding the hydrogen atoms with keeping the same unit cell parameter and fixing metal position.

As per the reviewer's helpful suggestion, we have included the voids ratio and pore structures in the revised Supplementary Information (page S18) as shown below.

Supplementary Table 4. The calculated void ratio in phase α , β and γ of PCP-3,5-pdc (probe radius: 1.2 Å)

	phase α	phase β	phase γ
Void ratio	15.4%	3.8%	19.8%

Supplementary Figure 8. The voids in phase α , β and γ of PCP-3,5-pdc (Highlighted in yellow, probe radius: 1.2 Å)

7. About the Hill coefficient, reference should be cited, and the discussion seems to be too simple and meaningless.

Response

We thank the reviewer for this comment. We have added the explanation and implication of Hill analysis in the Supplementary Information (Page S4-5) as follows.

The Hill coefficient, n , is recognized as an indicator of cooperative interactions because it describes the number of molecules bound per receptor^{2,3}. The Hill coefficient is the slope of the Hill plot $\{\log [Y/(1-Y)]$ versus $\log P\}$, where Y-axis is the CO₂ fractional unloading and P is the gas pressure^{4,5}. Generally, $n < 1$ corresponds to negatively cooperative systems, while $n > 1$ corresponds to positively cooperative systems. To evaluate the degree of cooperativity for the gate-opening step in CO₂ adsorption, we applied Hill's model analysis to the CO₂ adsorption isotherm at 195 K and 298 K.

The Hill coefficient for the gate-opening step in the CO₂ adsorption isotherm at 195 K was determined to be 4.1 (>1) (Supplementary Figure 9), confirming the positive, cooperative adsorption phenomenon. The positive CO₂ adsorption cooperativity in the structural transformation step was even stronger (Hill coefficient $n = 6.6$, Supplementary Figure 14). The difference in cooperativity at different temperature may

be due to the varying diffusion and stabilizing abilities of the CO₂ molecules (page 9 in the revised manuscript).

8. ...CO₂ required activation energy to adsorb onto... What's the meaning of activation energy? ...such adsorption did not occur... Which is "such adsorption"?

Response

We thank the reviewer for this comment. Activation energy means CO₂ molecules cannot freely enter the pore of PCP-3,5-pdc but has to overcome an energy barrier. After rechecking the caption of Supplementary Figure 13 (page S23), we removed the statement "The high-pressure gas sorption results and adsorption isotherms measured at low temperatures showed that CO₂ required activation energy to adsorb onto PCP-3,5-pdc. But such adsorption did not occur with C₂H₄, C₂H₂."

9. Specify the pressure of isobars. Specify the phase used for computational simulations.

Response

We thank the reviewer for this comment. The pressure for the isobar measurement is 100 kPa. We have added this information to the revised Supplementary Information (page S2). Additionally, both the activated and open phases were used in the computational simulations, and we have added this statement to the **Computational Details** section in the revised Supplementary Information (page S6).

10. The volume of N₂, CH₄, and C₂H₄ adsorbed in the binary adsorption mixtures was negligible. They should not be claimed as negligible. If negligible, the uptakes should not be compared and calculated. ... optimum selectivity... How optimum? This is a typical feature of gate-opening adsorption. What is "this"?

Response

We thank the reviewer for this comment.

We have revised the sentence "The volume of N₂, CH₄, and C₂H₄ adsorbed in the binary adsorption mixtures was negligible" to "The volume of N₂, CH₄, and C₂H₄ adsorbed in the binary adsorption mixtures was 0.5, 2.8 and 3.3 mL·g⁻¹, respectively" (page 11 in the revised manuscript).

Additionally, we have removed the word "optimum".

In reference 39, we have shown that simultaneous adsorption of more than one gas molecule is necessary to overcome the deformation energy of PCP framework induced by gas adsorption, which is an important feature for gate-opening adsorption. Therefore, we have revised the sentence "Interestingly, the total BE value is the largest (the most negative) when one CO₂ molecule is adsorbed at site I, two CO₂ molecules are adsorbed at site IIA, and one CO₂ molecule is adsorbed at one site IIB of two unit cells, which PCP·3.5CO₂ represents. This is a typical feature of gate-opening adsorption." to "Interestingly, the total BE value is the largest (the most negative) when one CO₂ molecule is adsorbed at site I, two CO₂ molecules are adsorbed at site IIA, and one CO₂

molecule is adsorbed at one site IIB of two unit cells, suggesting that simultaneous adsorption of more than one CO₂ molecules can happen in PCP-3,5-pdc, which is a typical feature of gate-opening adsorption” (Page 15 in the revised manuscript).

Reference:

1. Vervoorts, P. et al. Coordinated Water as New Binding Sites for the Separation of Light Hydrocarbons in Metal–Organic Frameworks with Open Metal Sites. *ACS. Appl. Mater. Inter.* **12**, 9448-9456 (2020).
2. Swenson, H. & Stadie, N.P. Langmuir’s theory of adsorption: A centennial review. *Langmuir.* **35**, 5409-5426 (2019).
3. Weiss, J.N. The Hill equation revisited: uses and misuses. *FASEB. J.* **11**, 835-841 (1997).
4. Wang, W., Wang, L., Huang, Y., Xie, Z. & Jing, X. Nanoscale Metal–Organic Framework–Hemoglobin Conjugates. *Chem. Asia. J.* **11**, 750-756 (2016).
5. Sen, S. et al. Cooperative Bond Scission in a Soft Porous Crystal Enables Discriminatory Gate Opening for Ethylene over Ethane. *J. Am. Chem. Soc.* **139**, 18313-18321 (2017).
6. Wang, L. et al. Designed metal-organic frameworks with potential for multi-component hydrocarbon separation. *Coord. Chem. Rev.* **484**, 215111 (2023).
7. Zhou, D.-D. & Zhang, J.-P. On the Role of Flexibility for Adsorptive Separation. *Acc. Chem. Res.* **55**, 2966-2977 (2022).
8. Chen, K.-J. et al. Tuning Pore Size in Square-Lattice Coordination Networks for Size-Selective Sieving of CO₂. *Angew. Chem. Int. Ed.* **55**, 10268-10272 (2016).
9. Shi, Y., Xie, Y., Alshahrani, T. & Chen, B. A zirconium-based microporous metal–organic framework for molecular sieving CO₂ separation. *Cryst. Eng. Commun.* **25**, 1643-1647 (2023).
10. Bloch, W.M., Babarao, R., Hill, M.R., Doonan, C.J. & Sumbly, C.J. Post-synthetic Structural Processing in a Metal–Organic Framework Material as a Mechanism for Exceptional CO₂/N₂ Selectivity. *J. Am. Chem. Soc.* **135**, 10441-10448 (2013).
11. Chen, F. et al. Carbon dioxide capture in gallate-based metal-organic frameworks. *Sep. Purif. Technol.* **292**, 121031 (2022).
12. Cui, J. et al. Kinetic-Sieving of Carbon Dioxide from Acetylene through a Novel Sulfonic Ultramicroporous Material. *Angew. Chem. Int. Ed.* **61**, e202208756 (2022).
13. Li, J.-R. et al. Porous materials with pre-designed single-molecule traps for CO₂ selective adsorption. *Nat. Commun.* **4**, 1538 (2013).
14. Yu, D., Yazaydin, A.O., Lane, J.R., Dietzel, P.D.C. & Snurr, R.Q. A combined experimental and quantum chemical study of CO₂ adsorption in the metal–organic framework CPO-27 with different metals. *Chem. Sci.* **4**, 3544-3556 (2013).
15. Zhou, D.-D. et al. A flexible porous Cu(ii) bis-imidazolate framework with ultrahigh concentration of active sites for efficient and recyclable CO₂ capture. *Chem. Commun.* **49**, 11728-11730 (2013).
16. Qazvini, O.T., Babarao, R. & Telfer, S.G. Selective capture of carbon dioxide from hydrocarbons using a metal-organic framework. *Nat. Commun.* **12**, 197 (2021).
17. Du, M. et al. Divergent Kinetic and Thermodynamic Hydration of a Porous Cu(II) Coordination Polymer with Exclusive CO₂ Sorption Selectivity. *J. Am. Chem. Soc.* **136**, 10906-10909 (2014).

18. Lee, W.R. et al. Diamine-functionalized metal–organic framework: exceptionally high CO₂ capacities from ambient air and flue gas, ultrafast CO₂ uptake rate, and adsorption mechanism. *Energ. Environ. Sci.* **7**, 744-751 (2014).
19. McDonald, T.M. et al. Capture of Carbon Dioxide from Air and Flue Gas in the Alkylamine-Appended Metal–Organic Framework mmen-Mg₂(dobpdc). *J. Am. Chem. Soc.* **134**, 7056-7065 (2012).
20. Kim, E.J. et al. Cooperative carbon capture and steam regeneration with tetraamine-appended metal-organic frameworks. *Science* **369**, 392-396 (2020).
21. Shekhah, O. et al. Made-to-order metal-organic frameworks for trace carbon dioxide removal and air capture. *Nat. Commun.* **5**, 4228 (2014).
22. Nugent, P. et al. Porous materials with optimal adsorption thermodynamics and kinetics for CO₂ separation. *Nature* **495**, 80-84 (2013).
23. Lin, R.-B., Chen, D., Lin, Y.-Y., Zhang, J.-P. & Chen, X.-M. A Zeolite-Like Zinc Triazolate Framework with High Gas Adsorption and Separation Performance. *Inorg. Chem.* **51**, 9950-9955 (2012).
24. Taylor, M.K. et al. Near-Perfect CO₂/CH₄ Selectivity Achieved through Reversible Guest Templating in the Flexible Metal–Organic Framework Co(bdp). *J. Am. Chem. Soc.* **140**, 10324-10331 (2018).
25. Horike, S. et al. Dense Coordination Network Capable of Selective CO₂ Capture from C₁ and C₂ Hydrocarbons. *J. Am. Chem. Soc.* **134**, 9852-9855 (2012).
26. Foo, M.L. et al. An Adsorbate Discriminatory Gate Effect in a Flexible Porous Coordination Polymer for Selective Adsorption of CO₂ over C₂H₂. *J. Am. Chem. Soc.* **138**, 3022-3030 (2016).

Reviewers' Comments:

Reviewer #1:

Remarks to the Author:

In my previous comments, I mentioned that the experimental details and methods for high-pressure experiments such as high-pressure sorption, high-pressure co-adsorption measurements, and high-pressure breakthrough experiments were not provided in sufficient detail. I believe that the author has included sufficient explanations to address my concerns. Additionally, as I read the rebuttal letter, I think that the author also adequately answered the majority of concerns comprehensively raised by other reviewers. Therefore, I would like to recommend publishing this paper after trivial revisions.

1. The CO₂/C₂H₄ breakthrough curves shown in Fig. S21 do not clearly indicate that PCP-3,5-pdc can effectively separate CO₂ from a CO₂/C₂H₄ binary mixture under dynamic conditions. The breakthrough curve shape of CO₂/C₂H₄ is similar to that of CO₂/N₂ before activation, which displayed poor separation performance due to humidity exposure. (as seen in Fig. S25) Based on the adsorption capacity of CO₂ and C₂H₄ at 10 bar and 298 K (condition of CO₂/C₂H₄ breakthrough curve feed gas) using the co-adsorption apparatus in Figure 3c, and the high-pressure isotherm in Figure S15, uptake of CO₂ is much higher than that of C₂H₄. (It seems like uptake of CO₂ is 4~5 times of C₂H₄) So, the retention time of CO₂ and C₂H₄ in the breakthrough curves should be significantly different but it seems like the breakthrough curve in Fig. S21 is not. I suggest quantifying the amount of gas uptake from the breakthrough curve and comparing it with the mixture of gas co-adsorption data and high-pressure isotherm. If there is a significant difference between the results, the authors should explain why PCP-3,5-PDC cannot work in dynamic conditions.

Reviewer #2:

Remarks to the Author:

I want to thank the authors for the extensive revisions and think they have addressed all comments adequately. The revised version is suitable for publication in Nature Comms.

Reviewer #3:

Remarks to the Author:

The revision addressed my concerns and can be accepted as it is.

Response to the reviewer comments

Journal Name: **Nature communications**

Manuscript Title: **Soft Corrugated Channel with Synergistic Exclusive Discrimination Gating for CO₂ Recognition in Gas Mixture**

Manuscript No: **NCOMMS-23-07080A**

Thank the referees for their helpful comments and suggestions. Based on these comments and suggestions, related changes have been done in the revised manuscript as follows:

Reviewer #1: In my previous comments, I mentioned that the experimental details and methods for high-pressure experiments such as high-pressure sorption, high-pressure co-adsorption measurements, and high-pressure breakthrough experiments were not provided in sufficient detail. I believe that the author has included sufficient explanations to address my concerns. Additionally, as I read the rebuttal letter, I think that the author also adequately answered the majority of concerns comprehensively raised by other reviewers. Therefore, I would like to recommend publishing this paper after trivial revisions.

1. The CO₂/C₂H₄ breakthrough curves shown in Fig. S21 do not clearly indicate that PCP-3,5-pdc can effectively separate CO₂ from a CO₂/C₂H₄ binary mixture under dynamic conditions. The breakthrough curve shape of CO₂/C₂H₄ is similar to that of CO₂/N₂ before activation, which displayed poor separation performance due to humidity exposure. (as seen in Fig. S25) Based on the adsorption capacity of CO₂ and C₂H₄ at 10 bar and 298 K (condition of CO₂/C₂H₄ breakthrough curve feed gas) using the co-adsorption apparatus in Figure 3c, and the high-pressure isotherm in Figure S15, uptake of CO₂ is much higher than that of C₂H₄. (It seems like uptake of CO₂ is 4~5 times of C₂H₄) So, the retention time of CO₂ and C₂H₄ in the breakthrough curves should be significantly different but it seems like the breakthrough curve in Fig.S21 is not. I suggest quantifying the amount of gas uptake from the breakthrough curve and comparing it with the mixture of gas co-adsorption data and high-pressure isotherm. If there is a significant difference between the results, the authors should explain why PCP-3,5-PDC cannot work in dynamic conditions.

Response

We are grateful to the reviewer for dedicating their time and effort to evaluate our manuscript. We truly appreciate the reviewer's constructive comments aimed at improving the quality of our manuscript.

Compared with the breakthrough separation of CO₂/N₂ and CO₂/CH₄, the separation performance of a CO₂/C₂H₄ binary mixture in PCP-35-pdc is relatively weak. Unlike N₂ and CH₄, C₂H₄ adsorption exhibits a higher uptake than CO₂ before pore opening at room temperature, indicating that PCP-3,5-pdc demonstrates stronger interactions with C₂H₄ than CO₂ in its initial state (see Fig. 3c). This observation is consistent with the calculated binding energies of these gas molecules at different pore-opening states for this PCP. The preferential adsorption of C₂H₄ in the closed phase could potentially affect the separation performance of the CO₂/C₂H₄ mixture due to the insufficient

adsorption equilibrium under dynamic conditions, in contrast to co-adsorption experiments. We have included this information on Page 11 of the revised manuscript.

Reviewer #2: I want to thank the authors for the extensive revisions and think they have addressed all comments adequately. The revised version is suitable for publication in Nature Comms.

Response

We are grateful to the reviewer for their time and effort in evaluating our manuscript. We really appreciate the reviewer's constructive comments to improve the quality of our manuscript.

Reviewer #3: The revision addressed my concerns and can be accepted as it is.

Response

We are grateful to the reviewer for their time and effort in evaluating our manuscript. We really appreciate the reviewer's constructive comments to improve the quality of our manuscript.